# The microRNA cluster C19MC confers differentiation potential into trophoblast lineages upon human pluripotent stem cells

Norio Kobayashi[1], Hiroaki Okae[1✉], Hitoshi Hiura[2], Naoto Kubota[3], Eri H. Kobayashi[1], Shun Shibata[1], Akira Oike[1], Takeshi Hori [4], Chie Kikutake [3], Hirotaka Hamada [1], Hirokazu Kaji [4], Mikita Suyama [3], Marie-Line Bortolin-Cavaillé[5], Jérôme Cavaillé[5] & Takahiro Arima[1✉]

The first cell fate commitment during mammalian development is the specification of the inner cell mass and trophectoderm. This irreversible cell fate commitment should be epigenetically regulated, but the precise mechanism is largely unknown in humans. Here, we show that naïve human embryonic stem (hES) cells can transdifferentiate into trophoblast stem (hTS) cells, but primed hES cells cannot. Our transcriptome and methylome analyses reveal that a primate-specific miRNA cluster on chromosome 19 (C19MC) is active in naïve hES cells but epigenetically silenced in primed ones. Moreover, genome and epigenome editing using CRISPR/Cas systems demonstrate that C19MC is essential for hTS cell maintenance and C19MC-reactivated primed hES cells can give rise to hTS cells. Thus, we reveal that C19MC activation confers differentiation potential into trophoblast lineages on hES cells. Our findings are fundamental to understanding the epigenetic regulation of human early development and pluripotency.

[1] Department of Informative Genetics, Environment and Genome Research Center, Tohoku University Graduate School of Medicine, Sendai 980-8575, Japan. [2] Department of Bioscience, Faculty of Life Science, Tokyo University of Agriculture, Tokyo 156-8502, Japan. [3] Division of Bioinformatics, Medical Institute of Bioregulation, Kyushu University, Fukuoka 812-8582, Japan. [4] Department of Biomechanics, Institute of Biomaterials and Bioengineering, Tokyo Medical and Dental University, Tokyo 101-0062, Japan. [5] Molecular, Cellular and Developmental biology department (MCD), Centre de Biologie Intégrative (CBI), University of Toulouse, CNRS, UPS, 31062 Toulouse, France. ✉email: hiroaki.okae.b4@tohoku.ac.jp; tarima@med.tohoku.ac.jp

The first cell fate decision in mammals occurs when toti-potent blastomeres differentiate into either the inner cell mass (ICM) or trophectoderm (TE). The ICM further differentiates into the epiblast and primitive endoderm. The epiblast gives rise to the entire fetus, the primitive endoderm contributes to the yolk sac, and the TE generates the placenta[1,2]. In both humans and mice, embryonic stem (ES) and trophoblast stem (TS) cells have been derived from ICM and TE cells, respectively[3–6]. Mouse ES (mES) cells are the in vitro counterpart of the pre-implantation epiblast, which is defined as naïve pluripotency[7], and contribute only to the epiblast lineage upon blastocyst injection[8]. Although mES cells are already committed to the epiblast lineage, they can transdifferentiate into mouse TS (mTS) cells in vitro following induction of TE-specifying tran-scription factors[9–11]. This transdifferentiation system has greatly contributed to our understanding of the mechanisms underlying the specification of the TE lineage in mice.

Recent studies reveal that naïve human ES (hES) cells derived using three different protocols[12–14] can spontaneously differ-entiate into trophoblast stem-like (hTSL) cells[15–18]. These hTSL cells have similar proliferation and differentiation capacities and transcriptome and methylome profiles to human TS (hTS) cells. Interestingly, there has also been accumulating evidence sug-gesting that primed hES cells spontaneously differentiate into trophoblast lineage cells upon BMP4 treatment[19]. Many researchers have utilized primed hES cells to study the develop-ment and function of the TE lineage, but there are skeptical views on these studies. Primed hES cells are most closely similar to late post-implantation epiblast cells[20], and post-implantation epiblast cells are unlikely to contribute to the TE lineage[21]. It's also been reported that BMP signaling induces differentiation of primed hES cells into mesoderm or amnion cells, not trophoblast cells[17,18,22,23]. Therefore, it is still controversial whether primed hES cells can differentiate into trophoblast cells.

In this study, we show that naïve hES cells can differentiate into hTS cells, but primed hES cells cannot. Using CRISPR/Cas-based genome and epigenome editing, we further demonstrate that the epigenetic status of a primate-specific miRNA cluster on chro-mosome 19 (C19MC) is the major determinant of the phenotypic difference between naïve and primed hES cells.

## Results

**Derivation of hTSL cells from naïve and primed hES cells**. We generated naïve hES cells from primed hES cells using N2B27 medium supplemented with 5i/L/A[14] (Supplementary Fig. 1a, b) and cultured them in hTS medium[6] after transient exposure to N2B27 medium without 5i/L/A. Naïve hES cells rapidly changed their morphology and differentiated into hTSL cells within a few passages (Fig. 1a). These hTSL cells are thereafter referred to as hTSL^naïve cells.

It has been reported that primed hES cells start to express markers of undifferentiated trophoblast cells within 48 h after BMP4 treatment, and the expression levels of these markers peak at day 2–5[24,25]. Thus, we treated primed hES cells with BMP4 for three days and then maintained them in hTS medium (Fig. 1a). Most cells displayed mesenchymal morphology after the BMP4 treatment, but hTSL cells gradually became dominant within several passages. These hTSL cells were designated as hTSL^primed cells (Fig. 1a). hTSL cells did not appear when primed hES cells were treated with BMP4 for six days or BMP4 pretreatment was omitted (Supplementary Fig. 1c). Thus, the duration and timing of BMP4 treatment seem to be crucial to derive hTSL^primed cells.

As with hTS cells, both hTSL^naïve and hTSL^primed cells expressed the TE-associated transcription factors GATA3, TFAP2C, TP63, ELF5, and TEAD4, but not the pluripotency

markers OCT4 and NANOG (Fig. 1b and Supplementary Fig. 1d). The transcription factor CDX2 was undetectable in hTSL^naïve and hTS^primed cells as in the case of hTS cells[6] (Supplementary Fig. 1d). Expression of the pan-TE marker KRT7, lack of HLA-ABC antigens, and hypomethylation of the ELF5 promoter are well-established markers of human trophoblast cells[26]. We confirmed that hTSL^naïve and hTSL^primed cells expressed KRT7 and did not express HLA-ABC (Fig. 1c,d). Moreover, the ELF5 promoter was hypomethylated in hTS, hTSL^naïve, and hTSL^primed cells but not in naïve and primed hES cells (Fig. 1e). These data suggest that hTSL^naïve and hTSL^primed cells are trophoblast cells.

**Proliferation and differentiation potentials of hTSL cells**. We previously reported that hTS cells can be propagated beyond the Hayflick limit of about 50 cell divisions, which is a hallmark of stem cells[6]. We confirmed that hTSL^naïve cells were expandable for at least 100 days, which corresponds to about 150 cell divi-sions (Fig. 2a). In contrast, hTSL^primed cells could not proliferate beyond the Hayflick limit and stopped dividing within 50 days (Fig. 2a). We next analyzed the differentiation capacity of hTSL^naïve and hTSL^primed cells. In the human placenta, there are two differentiated trophoblast lineages: extravillous cytotropho-blast (EVT) and syncytiotrophoblast (ST)[27,28]. Our previous study demonstrated that hTS cells can differentiate into EVT-like cells (EVT-hTS cells) through epithelial to mesenchymal transi-tion and ST-like cells (ST-hTS cells) via cell fusion[6]. We found that hTSL^naïve cells could differentiate into EVT-like cells as efficiently as hTS cells did, and the resultant cells expressed the EVT-specific marker HLA-G (Fig. 2b, c and Supplementary Movie 1, 2). Meanwhile, very few hTSL^primed cells differentiated into EVT-like cells (Fig. 2b and Supplementary Movie 3). We also revealed that hTSL^naïve cells could efficiently fuse to differentiate into ST-like cells, which expressed ST markers CGB and SDC1 and produced human chorionic gonadotropin (hCG) (Fig. 2d–f). Although some hTSL^primed cells also fused to form ST-like cells, their fusion efficiency was much lower than that of hTS or hTSL^naïve cells, and little hCG was secreted from ST-hTSL^primed cells (Fig. 2d–f). In summary, the proliferation and differentiation potentials of hTSL^primed cells were far inferior to those of hTS or hTSL^naïve cells.

**Transcriptome and DNA methylome profiling of hTSL cells**. To further characterize hTSL^naïve and hTSL^primed cells, we con-ducted RNA-seq (Supplementary Data 1). The gene expression profiles of hTSL^naïve and hTSL^primed cells were closest to those of hTS cells (Fig. 3a and Supplementary Fig. 2a). Likewise, ST-like cells derived from hTSL^naïve, hTSL^primed, and hTS cells exhibited high similarities. We also confirmed that EVT-like cells derived from hTSL^naïve and hTS cells had similar transcriptome profiles. We were unable to analyze EVT-like cells derived from hTSL^primed cells because such differentiation rarely occurred (Fig. 2b). We then focused on the expression levels of repre-sentative trophoblast markers[6,29], and confirmed that their expression levels were comparable among hTSL^naïve, hTSL^primed, and hTS cells both before and after differentiation (Supplemen-tary Fig. 2b). Although previous studies suggested that BMP4-treated primed hES cells differentiate into mesoderm or amnion cells[17,18,22,23], expression of mesoderm or amnion markers was negligible in hTSL^primed cells as in the case of hTSL^naïve and hTS cells (Supplementary Fig. 2b). These data reinforce the idea that hTSL^naïve and hTSL^primed are trophoblast cells.

Although hTSL^naïve, hTSL^primed, and hTS cells had similar gene expression patterns, hTSL^primed cells were phenotypically different from hTSL^naïve and hTS cells. To gain mechanistic insight into the poor proliferation and differentiation potentials

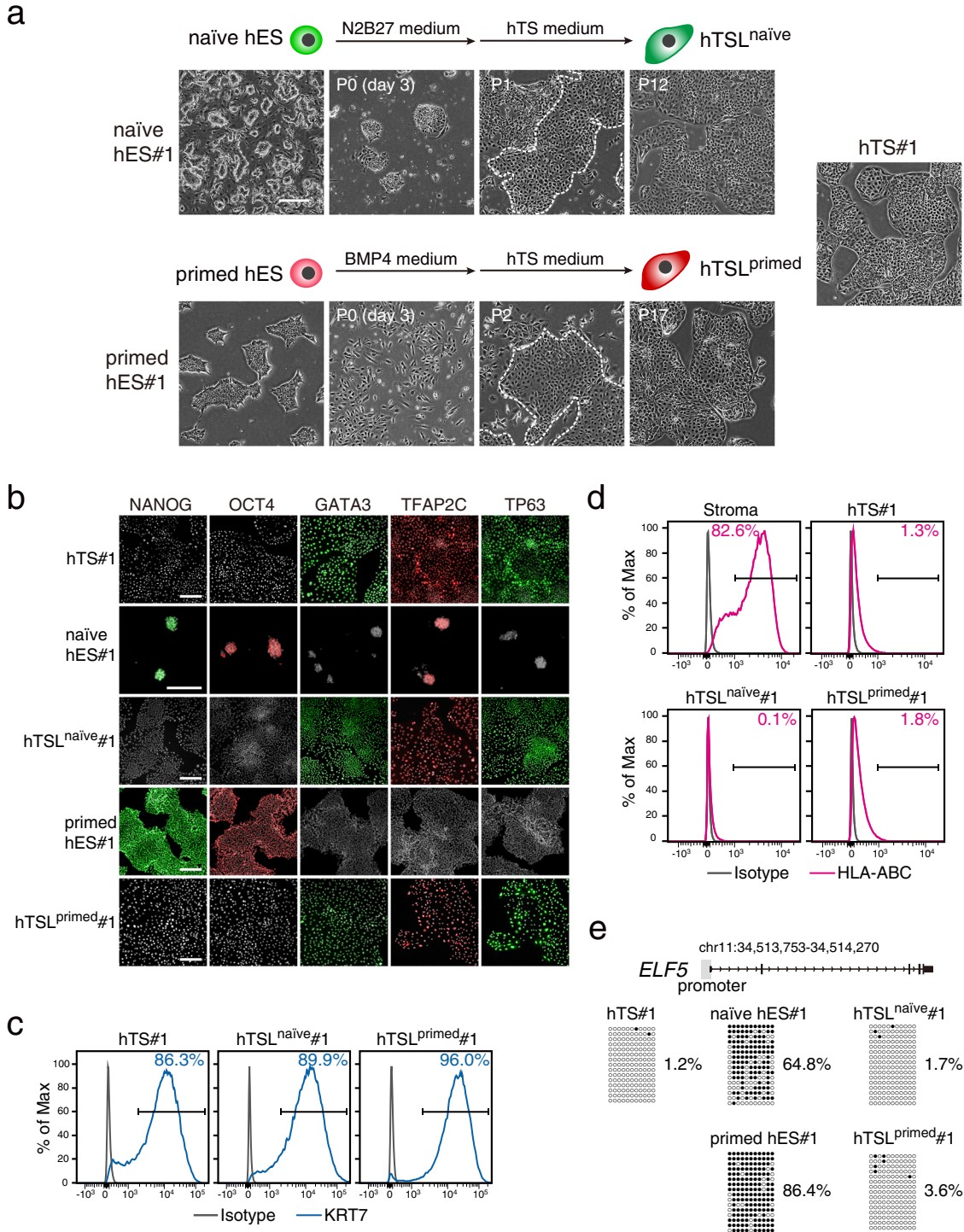

**Fig. 1 Derivation of hTSL cells from naïve and primed hES cells. a** Phase-contrast images of hTSL cells derived from naïve and primed hES cells. Naïve hES cells were derived from primed hES cells using 5i/L/A medium. After three days of culture in N2B27 medium, naïve hES cells were dissociated and cultured in hTS medium. Primed hES cells were treated with BMP4 for three days and then, the resultant cells were dissociated and cultured in hTS medium. hTSL cells were visible within a few passages (white dotted line). Similar results were obtained with two independent hES cell lines. An image of hTS cells (hTS#1) is shown for comparison. The scale bar indicates 300 μm. **b** Immunostaining of pluripotency markers (NANOG and OCT4) and TE-associated transcription factors (GATA3, TFAP2C, and TP63) in hES, hTS, and hTSL cells. Nuclei were stained with Hoechst 33,258 (shown in gray). The scale bar indicates 200 μm. Similar results were obtained with two independent cell lines. **c** Flow cytometry analysis of KRT7 expression in hTSL cells. Representative data obtained from two independent cell lines are shown. hTS cells were analyzed as a positive control. **d** Flow cytometry analysis of HLA-ABC expression on hTSL cells. Representative data obtained from two independent cell lines are shown. Stromal cells isolated from human placentas were analyzed as a positive control. **e** DNA methylation analysis of the *ELF5* promoter by bisulfite sequencing. Black and white circles indicate methylated and unmethylated CpGs, respectively. The methylation levels are shown on the right. The analyzed region is highlighted by a grey rectangle.

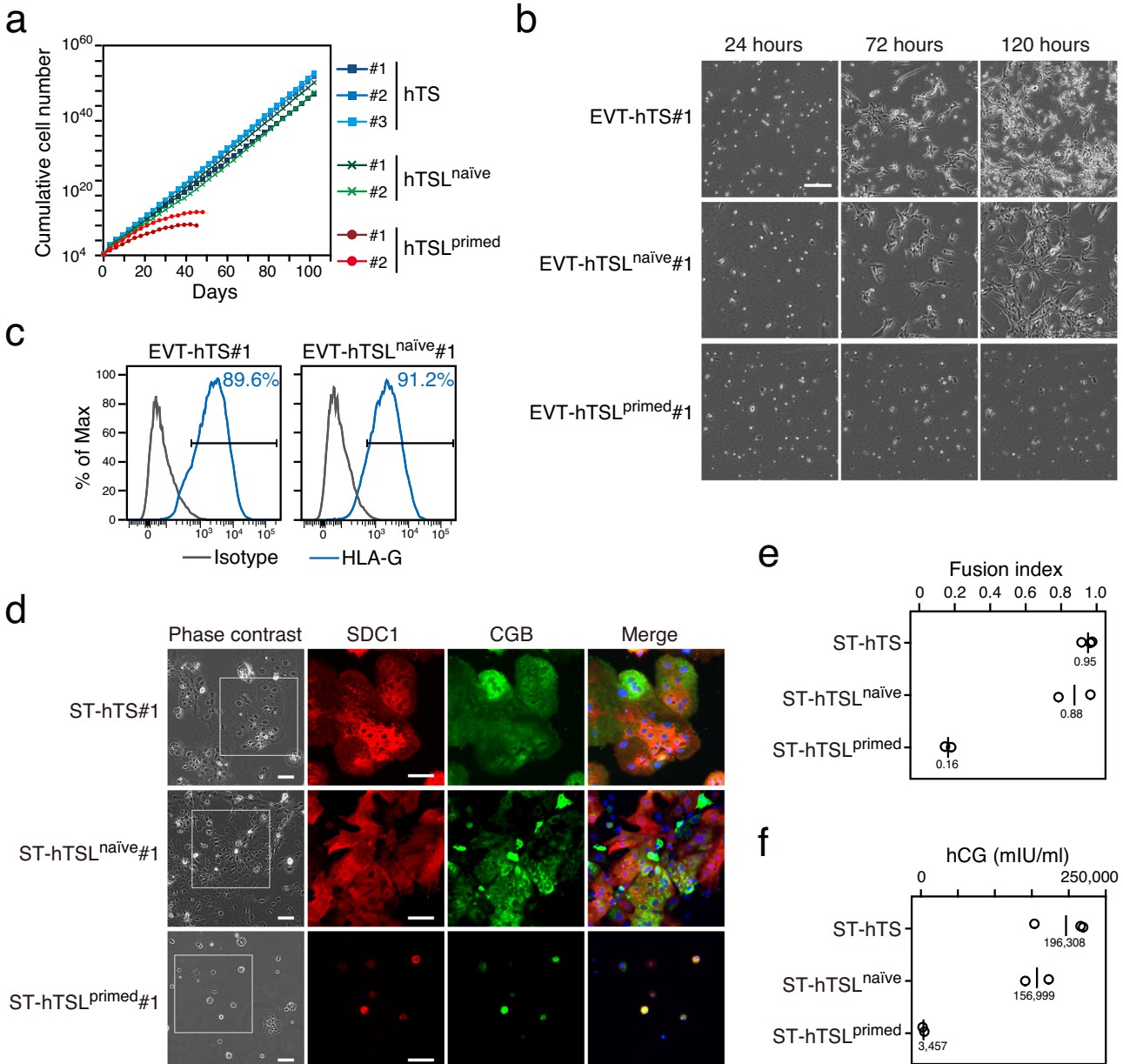

**Fig. 2 Proliferation and differentiation potentials of hTSL cells. a** Growth curve of hTS and hTSL cells. Although hTS and hTSL^naïve cells could be stably maintained for more than 100 days, hTSL^primed cells stopped proliferation within 50 days. **b** Time-lapse analysis of EVT-like cells differentiating from hTS and hTSL cells. The phase-contrast images are taken from time-lapse Supplementary Movie 1–3. The scale bar indicates 300 µm. Similar results were obtained with two independent cell lines. **c** Flow cytometry analysis of HLA-G expression on EVT-like cells derived from hTS and hTSL^naïve cells. Representative data obtained from two independent cell lines are shown. **d** Phase contrast and immunofluorescence images of ST-like cells derived from hTS and hTSL cells. Cells were stained for SDC1 and CGB. Nuclei were stained with Hoechst 33258. The scale bar indicates 100 µm. Similar results were obtained with two independent cell lines. **e** Fusion efficiency of ST-hTS and -hTSL cells. The fusion index is defined as (N-S)/T, where *N* is the number of nuclei in the syncytia, S is the number of syncytia, and T is the total number of nuclei counted. Three ST-hTS, two ST-hTSL^naïve, and two ST-hTSL^primed cell lines were analyzed. Black bars and numbers indicate the mean fusion indexes. **f** Levels of hCG secreted by ST-hTS and -hTSL cells. Three ST-hTS, two ST-hTSL^naïve, and two ST-hTSL^primed cell lines were analyzed. Black bars and numbers indicate the mean amount of hCG.

of hTSL^primed cells, we compared the transcriptome profile of hTSL^primed cells with those of hTS and hTSL^naïve cells. This analysis led to the identification of 274 down-regulated and 612 up-regulated genes (Supplementary Fig. 2c). Subsequent pathway enrichment analysis revealed that cell-cycle-related pathways were enriched in the down-regulated genes (Supplementary Fig. 2d), which was consistent with the poor proliferation potential of hTSL^primed cells. We also found that pathways involved in cell-extracellular matrix (ECM) interactions were enriched in the up-regulated genes (Supplementary Fig. 2d). ECM

plays crucial roles in EVT differentiation[30], and abnormal expression of ECM-related genes might provide an explanation of why hTSL^primed cells rarely differentiated into EVT-like cells.

Human trophoblast cells have unique DNA methylation features, such as global hypomethylation and placenta-specific germline differentially methylated regions (gDMRs)[6,31,32]. To examine whether these unique features are observed in hTSL cells, we performed whole-genome bisulfite sequencing (WGBS) of hTSL^primed cells. We also analyzed publicly available WGBS data of naïve hES[33], primed hES[34], hTS[6], and hTSL^naïve cells[15].

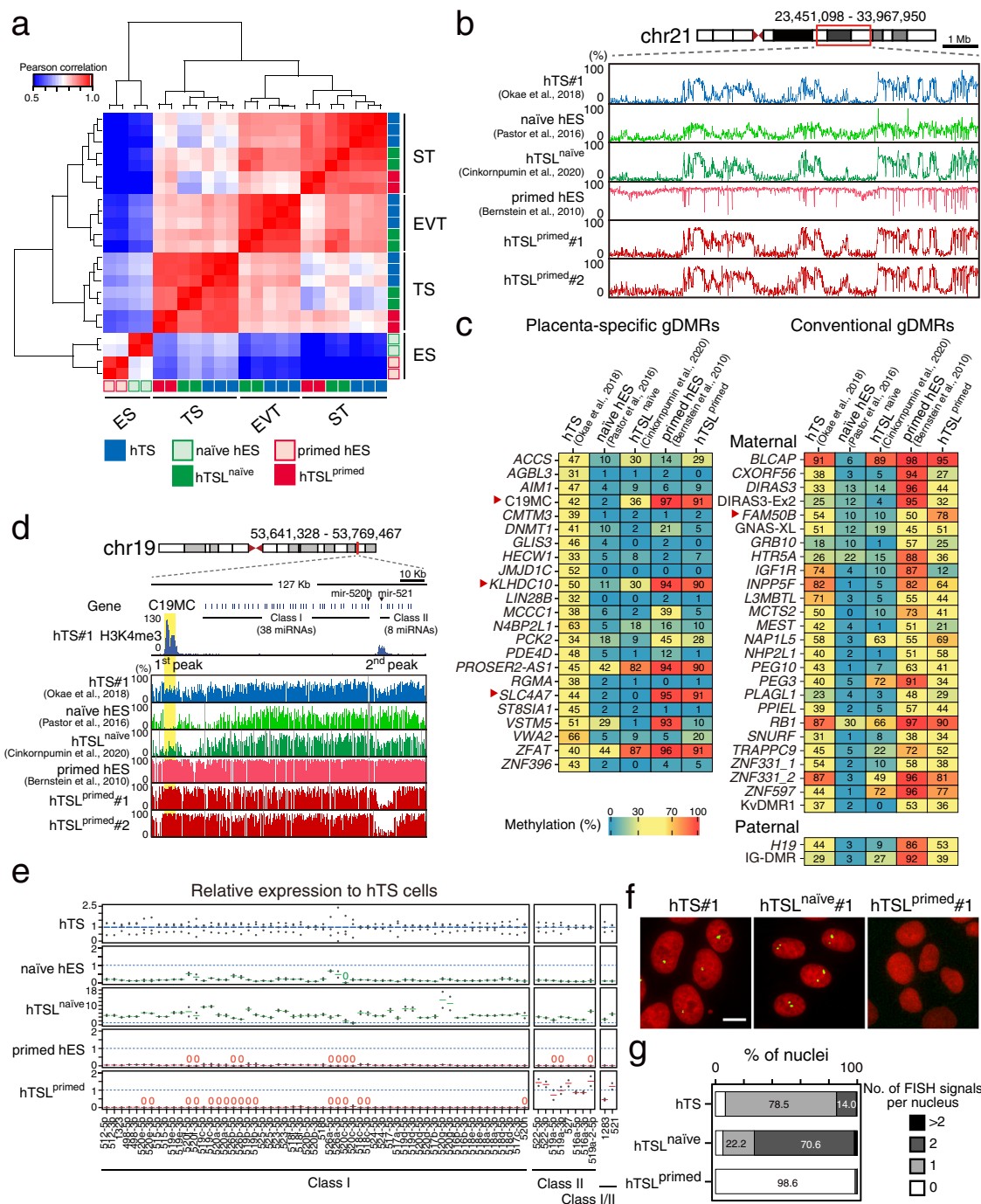

**Fig. 3 Transcriptome and DNA methylome profiling of hTSL cells. a** Heatmap representation of Pearson correlation coefficients between hES, hTS, and hTSL cells and their differentiated derivatives. Only 8768 differentially expressed genes (TPM > 4 in at least one cell type, adjusted *p*-value < 0.05, fold change > 4) were analyzed. hES, hTS, and hTSL cells are color-coded as indicated. **b** DNA methylation patterns in hES, hTS, and hTSL cells. A part of chromosome 21 is indicated. **c** Heatmap representation of DNA methylation levels of gDMRs in hES, hTS, and hTSL cells. Heatmaps for placenta-specific (left) and conventional (right) gDMRs are represented. All gDMRs, except for the *H19* DMR and IG-DMR, are maternally methylated. Four gDMRs that were hypermethylated in hTSL^primed cells but not in hTSL^naïve and hTS cells are indicated by red arrowheads. **d** H3K4me3-enriched peaks and DNA methylation patterns at the C19MC locus. The C19MC DMR is highlighted in yellow. Vertical bars indicate methylation levels and gray indicates missing data or regions without CpG sites. Please see Supplementary Fig. 3b for the DNA methylation patterns of the C19MC DMR in hTSL^naïve cells derived in this study. **e** Relative expression levels of C19MC miRNAs in hES, hTS, and hTSL cells. Expression levels in hTS cells are set as 1. Three hTS, two naïve hES, two hTSL^naïve, two primed hES, and two hTSL^primed cell lines were analyzed. Bar charts are shown as mean. Note that hsa-mir-1283 and -521 have copies in both the Class I and II regions. **f** RNA-FISH of C19MC in hTS and hTSL cells. C19MC expression was labeled with an Alexa Fluor 488-labeled oligonucleotide probe (shown in green). Nuclei were stained with Hoechst 33,258 (shown in red). The scale bar indicates 10 μm. **g** Percentage of the number of C19MC FISH signals per nucleus in hTS and hTSL cells. Three hTS, two hTSL^naïve, and two hTSL^primed cell lines were analyzed, and the count data were summed up for each cell type. 500–1000 nuclei were examined. Nuclei in M-phase were excluded from the analysis. Publicly available WGBS data were used for hTS cells[6], naïve hES cells[33], hTSL^naïve cells[15], and primed hES cells[34].

hTSL[naïve], hTSL[primed], and hTS cells had similar methylome profiles, characterized by large hyper- and hypomethylated domains (Fig. 3b and Supplementary Fig. 2e). Strikingly, the methylation level in hTSL[primed] cells was much lower than that in primed hES cells, suggesting that global DNA demethylation may occur during the transition from primed hES to hTSL[primed] cells. We further compared the methylation patterns of various genomic features, including CpG islands (CGI), promoters, gene bodies, retrotransposons, and gDMRs (Supplementary Fig. 2e). Although most genomic features showed high similarities among hTSL[naïve], hTSL[primed], and hTS cells (R > 0.89), gDMRs were the exception. We classified gDMRs as conventional or placenta-specific according to our previous study[6,32] and analyzed their methylation levels (Supplementary Data 2). As previously reported, almost all gDMRs were hypomethylated (methylation levels of <20%) in naïve hES cells (Fig. 3c)[33]. Similar patterns were also observed in hTSL[naïve] cells, although some gDMRs gained methylation during the transition from naïve hES to hTSL[naïve] cells. In primed hES cells, some conventional gDMRs maintained intermediate methylation levels (30–70%). However, the other conventional gDMRs were hypermethylated (methylation levels of >80%), and most placenta-specific gDMRs were hypo- or hypermethylated in primed hES cells. Such a tendency was conserved in hTSL[primed] cells (Fig. 3c and Supplementary Fig. 2e).

Genomic imprinting is essential for normal placental development[35], and we anticipated that abnormal imprinting might account for the poor proliferation and differentiation observed in hTSL[primed] cells. Considering that most gDMRs were hypomethylated in hTSL[naïve] cells but these cells still retain proliferation and differentiation potentials comparable to hTS cells, we focused on gDMRs that were hypermethylated in hTSL[primed] cells but not in hTSL[naïve] and hTS cells. The C19MC, KLHDC10, SLC4A7, and FAM50B DMRs met this criterion (Fig. 3c). Among them, we decided to further analyze C19MC because the expression levels of KLHDC10, SLC4A7, and FAM50B were comparable between hTSL[primed] and hTS cells (Supplementary Fig. 3a). The C19MC DMR is located on human chr19q13 and serves as the promoter of a ~100 kb-large primate-specific miRNA cluster that contains 46 miRNAs[36,37] (Fig. 3d). C19MC is maternally imprinted and paternally expressed, and its expression is almost restricted to the placenta[6,38]. We previously revealed that the allele-specific methylation of the C19MC DMR is maintained in hTS cells[6]. It should be noted that although the C19MC DMR was intermediately methylated in hTSL[naïve] cells generated by Cinkornpumin et al. (Fig. 3c, d)[15], this region was almost completely demethylated in our hTSL[naïve] cells (Supplementary Fig. 3b, c).

We next performed miRNA-seq to analyze the expression levels of C19MC miRNAs (Supplementary Data 3). We found that almost all of the C19MC miRNAs showed higher expression levels in hTSL[naïve] cells than in hTS cells, whereas most C19MC miRNAs, except for those located near the 3′ end of C19MC, were hardly expressed in hTSL[primed] cells (Fig. 3e). The expression pattern of C19MC miRNAs in hTSL[primed] cells implied that the C19MC DMR might not be the only promoter regulating this miRNA cluster. Thus, we performed ChIP-seq for H3K4me3, a marker of active promoters, and found that along with the C19MC DMR, there might be a novel promoter between hsa-mir-520h and -521 ('2nd peak' in Fig. 3d). Consistently, the miRNAs highly expressed in hTSL[primed] cells were all located downstream of the putative second promoter (Fig. 3e). Based on these results, we classified the C19MC miRNAs as follows: those between the C19MC DMR and the second putative promoter were designated as Class I and those located downstream of the second promoter as Class II. Hsa-mir-1283 and -521 have copies in both the Class I and II regions and were named Class I/II. To better understand the allelic expression patterns of C19MC, we also performed RNA fluorescence in situ hybridization (FISH) of C19MC (Fig. 3f, g). One FISH signal was detected in most hTS cells (78.5%), consistent with the imprinted expression of C19MC. Meanwhile, two FISH signals were detected in the majority of hTSL[naïve] cells (70.6%), and no FISH signal was detected in almost all hTSL[primed] cells (98.6%). These FISH signal patterns were consistent with the methylation levels of the C19MC DMR (Fig. 3d and Supplementary Fig. 3b, c). Taken together, we revealed that C19MC is active in hTSL[naïve] and hTS cells, whereas most C19MC miRNAs, except for Class II ones, are silenced in hTSL[primed] cells.

**Functional analysis of C19MC in hTS cells**. To determine whether C19MC is required for proliferation and differentiation of hTS cells, we deleted the C19MC DMR in hTS cells using the CRISPR/Cas system (Fig. 4a). We previously performed an allele-specific DNA methylation analysis of the parental hTS line (hTS#1) and identified several SNPs that can be used to distinguish the maternal and paternal alleles of the C19MC DMR[6]. We isolated four maternal (hTS[Δm/+]), one paternal (hTS[+/Δp]), and two homozygous deletion (hTS[Δm/Δp]) clones (Fig. 4b and Supplementary Fig. 4a, b). We confirmed that hTS[Δm/+] cells had only the hypomethylated paternal allele and hTS[+/Δp] cells had only the hypermethylated maternal allele (Fig. 4c and Supplementary Fig. 4c). The expression levels of Class I miRNAs were considerably lower in hTS[+/Δp] and hTS[Δm/Δp] cells than those in genetically unmodified hTS cells or hTS[Δm/+] cells (Fig. 4d). In contrast, the decrease in Class II miRNAs expression, which might be regulated by the putative second promoter as discussed above, was relatively modest in hTS[+/Δp] and hTS[Δm/Δp] cells. RNA-FISH of C19MC showed that most hTS[Δm/+] cells (84.6%) had one FISH signal and almost all hTS[+/Δp] and hTS[Δm/Δp] cells (97.9%) had no FISH signal (Fig. 4e, f). Therefore, the expression pattern of the C19MC miRNAs in hTS[+/Δp] and hTS[Δm/Δp] cells resembles that in hTSL[primed] cells.

As with hTSL[primed] cells, hTS[+/Δp] and hTS[Δm/Δp] cells exhibited extremely lower proliferation rates than genetically unmodified hTS cells or hTS[Δm/+] cells (Fig. 4g). It should be noted that hTS[+/Δp] and hTS[Δm/Δp] cells were propagated from single cells before analysis, which took about two months. Thus, the estimated life span of hTS[+/Δp] and hTS[Δm/Δp] cells was around 2–3 months, comparable to that of hTSL[primed] cells. We also found that differentiation into EVT-like cells was severely impaired in hTS[+/Δp] and hTS[Δm/Δp] cells (Fig. 4h). Although some ST-like cells were obtained from hTS[+/Δp] and hTS[Δm/Δp] cells, the fusion efficiency was low and these ST-like cells secreted only a small amount of hCG (Fig. 4h–j). Contrary to hTS[+/Δp] and hTS[Δm/Δp] cells, hTS[Δm/+] cells could efficiently differentiate into EVT- and ST-like cells. These data reveal that C19MC is essential for normal proliferation and differentiation of hTS cells and C19MC-deficient hTS cells phenocopy hTSL[primed] cells.

We next asked whether C19MC was also required for the derivation of highly proliferative bipotent hTSL cells from naïve hES cells as follows. We first deleted the C19MC DMR in primed hES cells and isolated two heterozygous (hES[+/Δ]) and two homozygous (hES[Δ/Δ]) deletion clones (Supplementary Fig. 4d). Then, these cells were subjected to 5i/L/A medium to generate naïve hES cells. Naïve hES cells were successfully derived from the hES[+/Δ] clones but not from the hES[Δ/Δ] clones (Supplementary Fig. 4e, f). We were therefore unable to directly address whether naïve hES[Δ/Δ] cells have potential for differentiation into trophoblast lineages. However, this finding raises the interesting possibility that C19MC may be required for the transition from the primed state to the naïve state.

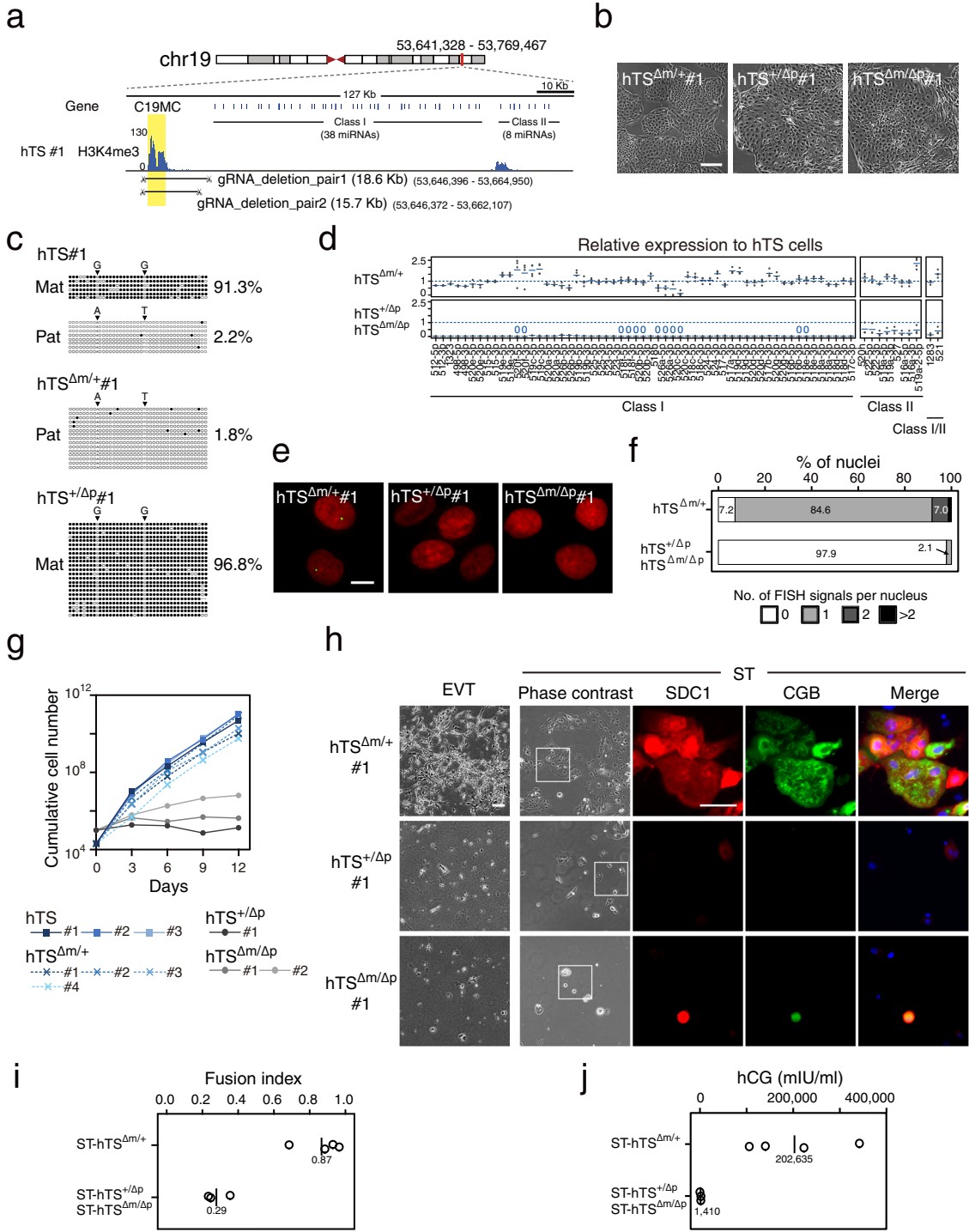

**Reactivation of C19MC in primed hES cells**. We next asked whether reactivation of C19MC could rescue the poor proliferation and differentiation potentials of hTSL^primed cells. We utilized the dCas9-peptide repeat and scFv-TET1 catalytic domain fusions (dCas-TET)[39] to demethylate the C19MC DMR (Supplementary Fig. 5a). The C19MC DMR contains short tandem repeats[38], and we designed and cloned a guide RNA (gRNA) targeting these repeats. We delivered this gRNA and the dCas-TET expression vectors into a primed hES cell line (primed hES#1) and derived hTSL cells, which were designated as hTSL^C19MC cells (Fig. 5a and Supplementary Fig. 5b, c). We also isolated two C19MC-positive clones from the bulk hTSL^C19MC cells by single-cell cloning. As a negative control, we used an empty gRNA vector

and the derived hTSL cells were named hTSL^empty cells. Single-cell cloning was not applicable to hTSL^empty cells due to their poor proliferation potential and we analyzed only bulk hTSL^empty cells.

RNA-FISH of C19MC showed that 62.3% of bulk hTSL^C19MC cells had one or two FISH signal(s) and the proportion exceeded 85% in the hTSL^C19MC clones (Fig. 5b, c). Consistently, the C19MC DMR was almost completely demethylated and Class I miRNAs were highly expressed in hTSL^C19MC clones (Fig. 5d, e). By contrast, the C19MC DMR was hypermethylated in hTSL^empty cells. We found that hTSL^C19MC clones showed high proliferation rates comparable to hTS cells, whereas hTSL^empty cells grew much slower (Fig. 5f). Moreover, hTSL^C19MC clones had the

**Fig. 4 Deletion of the C19MC DMR in hTS cells. a** Location of the gRNAs used to delete the C19MC DMR. The C19MC DMR was deleted in hTS cells using two pairs of gRNAs (gRNA_deletion_pair1 and 2). **b** Phase-contrast images of hTS cell lines with deletion of the C19MC DMR. We obtained four maternal (hTS$^{\Delta m/+}$), one paternal (hTS$^{+/\Delta p}$), and two homozygous deletion (hTS$^{\Delta m/\Delta p}$) clones, and representative images are shown. The scale bar indicates 300 μm. **c** DNA methylation analysis of the C19MC DMR by bisulfite sequencing. Black and white circles indicate methylated and unmethylated CpGs, respectively. The methylation levels are shown on the right. SNPs are shown by arrowheads. The DNA methylation patterns of the other cell lines are presented in Supplementary Fig. 4c. **d** Relative expression levels of C19MC miRNAs in C19MC knockout hTS cells. Expression levels in hTS cells are set as 1. Four hTS$^{\Delta m/+}$, one hTS$^{+/\Delta p}$, and two hTS$^{\Delta m/\Delta p}$ clones were analyzed. Bar charts are shown as mean. **e** RNA-FISH of C19MC in hTS cells with deletion of the C19MC DMR. C19MC expression is shown in green and nuclei are in red. The scale bar indicates 10 μm. **f** Percentage of the number of C19MC FISH signals per nucleus in hTS cells with deletion of the C19MC DMR. Four hTS$^{\Delta m/+}$ cell lines and three hTS$^{+/\Delta p}$ or hTS$^{\Delta m/\Delta p}$ cell lines were analyzed and the count data were summed up. 500–1000 nuclei were examined for each cell type. **g** Growth curve of hTS cells with deletion of the C19MC DMR. Cells were cultured for 12 days. Genetically unmodified hTS cells were also analyzed for comparison. **h** Phase-contrast images of EVT- and ST-like cells derived from hTS cells with deletion of the C19MC DMR. ST-like cells were immunostained for SDC1 and CGB. Nuclei were stained with Hoechst 33,258. The scale bar indicates 100 μm. Four hTS$^{\Delta m/+}$, one hTS$^{+/\Delta p}$, and two hTS$^{\Delta m/\Delta p}$ clones were analyzed, and representative data are shown. **i** Fusion efficiency of ST-like cells derived from hTS cells with deletion of the C19MC DMR. Black bars and numbers indicate mean fusion indexes. **j** Levels of hCG secreted by ST-like cells derived from hTS cells with deletion of the C19MC DMR. Black bars and numbers indicate the mean amount of hCG.

capacity to efficiently differentiate into EVT- and ST-like cells, but again, hTSL$^{empty}$ cells did not (Fig. 5g–j). We also performed RNA-seq and confirmed that hTSL$^{C19MC}$ cells had transcriptome profiles similar to hTS and hTSL$^{naïve}$ cells both before and after differentiation (Supplementary Fig. 5d). However, in spite of the overall similarities, we noticed that ST- hTSL$^{C19MC}$ cells secreted three to four times less hCG than ST-hTS or ST-hTSL$^{naïve}$ cells (Figs. 2f and 5k). Furthermore, when compared with ST-hTS and ST-hTSL$^{naïve}$ cells, cell-cycle-related genes tended to be up-regulated in ST-hTSL$^{C19MC}$ cells (Supplementary Fig. 5e, f). Although the underlying mechanism was unclear, these observations suggested insufficient maturation of ST-hTSL$^{C19MC}$ cells because mature ST cells are terminally differentiated and characterized by cell cycle arrest. Taken together, we revealed that highly proliferative bipotent hTSL cells can be derived from C19MC-reactivated primed hES cells, although ST-like cells derived from these hTSL cells are rather immature. From these data, we concluded that the silencing of C19MC is the major, if not sole, barrier that prevents differentiation of primed hES cells into hTS cells.

We also applied the dCas-TET system to hTSL$^{primed}$ cells that were derived from primed hES cells without genetic manipulation (Supplementary Fig. 5b, g). We confirmed that 35.9% of the resultant cells, designated as hTSL$^{primed-C19MC}$ cells, were C19MC-positive (Supplementary Fig. 5h, i). However, to our surprise, these cells rarely differentiated into EVT- or ST-like cells and their proliferation potential was too low to apply single-cell cloning (Supplementary Fig. 5j). Therefore, C19MC reactivation after the derivation of hTSL$^{primed}$ cells could not restore their proliferation and differentiation potentials.

**Predicting target genes of the C19MC miRNAs.** To better understand how C19MC functions, we performed a computational target prediction of Class I miRNAs using mirDIP[40] and identified 4734 genes. We confirmed that these genes showed significantly higher expression levels in C19MC inactive (hTSL$^{primed}$, hTS$^{+/\Delta p}$, hTS$^{\Delta m/\Delta p}$, and hTSL$^{empty}$) cells than in C19MC active (hTS, hTSL$^{naïve}$, hTS$^{\Delta m/+}$, and hTSL$^{C19MC}$) cells (Supplementary Fig. 6a). Among these 4734 genes, 310 genes were up-regulated >1.5-fold in C19MC inactive cells and regarded as the most likely candidates (Supplementary Fig. 6b). Subsequent pathway enrichment analysis revealed that p53-related pathways were significantly enriched among these 310 genes (Supplementary Fig. 6c), which was consistent with the poor proliferation potential of the C19MC inactive cells. We further looked at the combination between the p53 signaling-associated genes and Class I miRNAs and found that some negative regulators of cell growth, such as *CCNG2*[41], *CDKN1A*[42–44], *PMAIP1*[45],

*TP53INP1*[46,47], and *ZMAT3*[48], were potentially targeted by multiple C19MC miRNAs (Supplementary Fig. 6d). In addition to p53-related pathways, ECM-related pathways were also enriched in the 310 genes. As already discussed above, given the importance of ECM in EVT differentiation, aberrant expression of ECM-related genes might be one reason why C19MC inactive cells rarely differentiated into EVT-like cells.

**Discussion**

There has been a long-standing debate on whether BMP4 induces differentiation of primed hES cells into trophoblast cells[21,23,49]. Although our hTSL$^{primed}$ cells have poor proliferation and differentiation potentials, their transcriptome and methylome profiles strongly suggest that they are trophoblast cells. Thus, our data support the idea that primed hES cells can differentiate into trophoblast cells in a broad sense. What remains unclear, however, is whether such differentiation is just an artifact of in vitro culture or can occur in vivo. Our data are favorable to the former due to the following reasons. First, C19MC miRNA expression is a hallmark of human trophoblast cells[26], but hTSL$^{primed}$ cells do not express most C19MC miRNAs. Second, the lifespan of hTSL$^{primed}$ cells is much shorter than the human pregnancy period. Finally, although EVT and ST cells are essential for normal placental function, hTSL$^{primed}$ cells rarely differentiate into EVT-like cells and their differentiation into ST-like cells is also compromised. In addition, it is notable that although recent studies on primate post-implantation embryos and in vitro-cultured human embryos have greatly advanced our understanding of human post-implantation development, there has been no evidence that human post-implantation epiblast cells contribute to the TE lineage[20,50].

Recently, Li et al. and Mischler et al. reported the derivation of hTSL$^{primed}$ cells that can differentiate into both EVT- and ST-like cells, which is inconsistent with our findings[51,52]. Although both Li et al. and Mischler et al. maintained hTSL$^{primed}$ cells in hTS medium that we had developed, their protocols used for the pretreatment of primed hES cells are substantially different from ours. Li et al. generated trophoblast cysts using a micromesh culture technique. Mischler et al. induced and expanded trophoblast cells using media containing sphingosine-1 phosphate or its agonist. Such differences might explain the phenotypic differences between our hTSL$^{primed}$ cells and those generated by Li et al. and Mischler et al. However, Li et al. and Mischler et al. did not analyze C19MC or make a direct comparison between hTS and their hTSL$^{primed}$ cells. Therefore, it is unclear whether hTSL$^{primed}$ cells generated by Li et al. and Mischler et al. have equivalent properties to hTS cells.

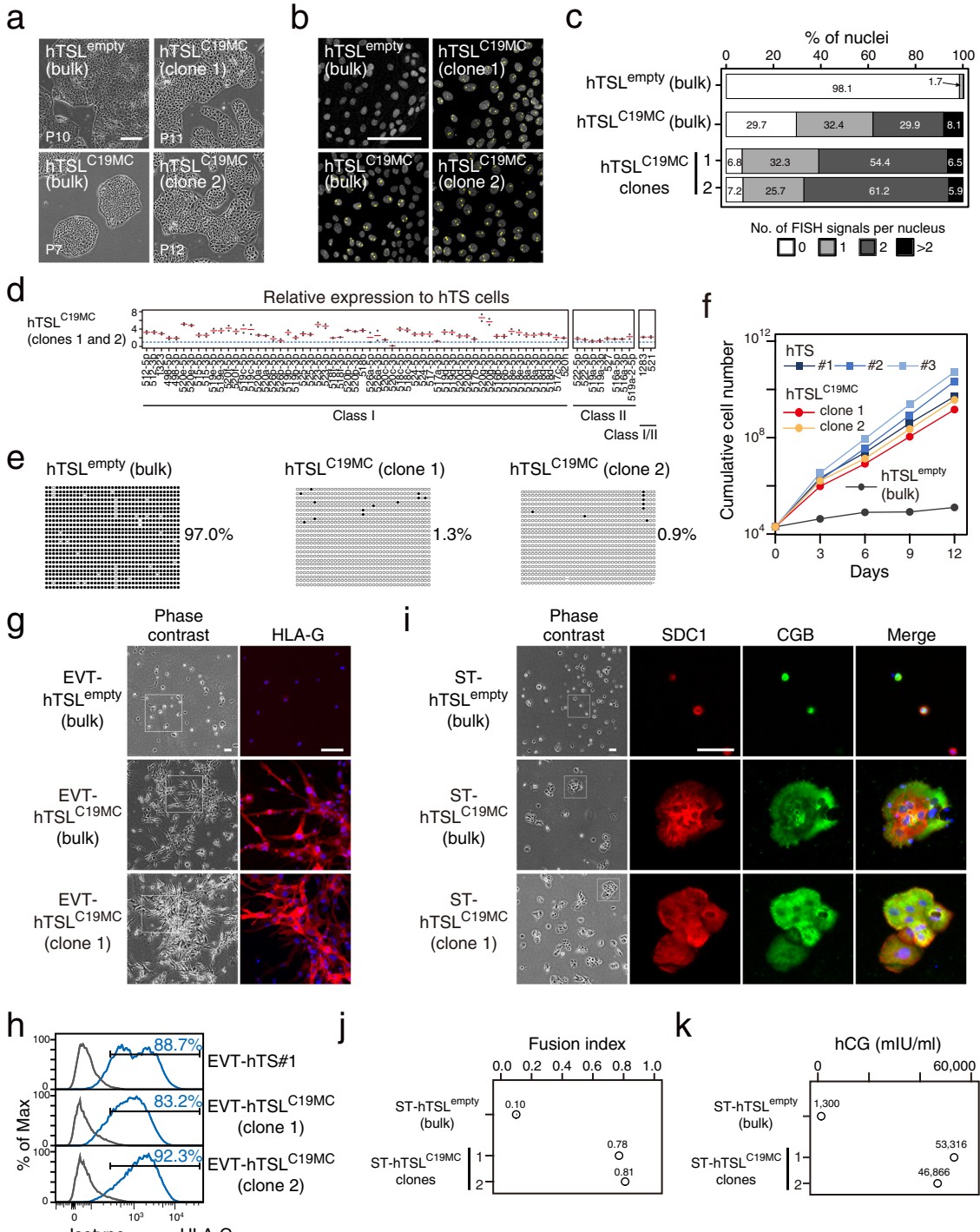

Our findings on the C19MC DMR are analogous to those on the mouse *Elf5* promoter[53]. Ng et al. demonstrated that the *Elf5* promoter is hypermethylated in mES cells and its demethylation by *Dnmt1* knockout triggers ectopic differentiation of trophoblast cells both in vivo and in vitro. We revealed that the *ELF5* promoter is also hypermethylated in primed hES cells but almost completely demethylated in hTSL^primed cells. This observation strongly suggests that the poor proliferation and differentiation potentials of hTSL^primed cells cannot be explained by the epigenetic status of the *ELF5* promoter. Alternatively, we uncovered that hypermethylation of the C19MC DMR is responsible for these abnormalities observed in hTSL^primed cells (Fig. 6). Therefore, at least in our in vitro differentiation system, the methylation

status of the C19MC DMR is more robust than that of the *ELF5* promoter. Although it remains unknown when the C19MC DMR undergoes hypermethylation in the epiblast lineage during human development, we speculate that the C19MC DMR might contribute to the irreversible segregation of the epiblast and trophoblast lineages in humans as the *Elf5* promoter does in mice.

We showed that highly proliferative bipotent hTSL cells can be derived from C19MC-reactivated primed hES cells (Fig. 6). However, paradoxically, C19MC reactivation in hTSL^primed cells could not rescue their poor proliferation and differentiation potentials. There are at least two possibilities to explain this discrepancy: (1) C19MC might be required not only for the maintenance of hTS cells but also for the transition from primed

**Fig. 5 Derivation of highly proliferative bipotent hTSL cells from C19MC-reactivated primed hES cells. a** Phase-contrast images of hTSL cells obtained from C19MC-reactivated primed hES cells (hTSL$^{C19MC}$ cells). We also analyzed two clones isolated from these bulk hTSL$^{C19MC}$ cells. hTSL$^{empty}$ cells were derived from hES cells transfected with an empty gRNA expression vector and used for comparison. The scale bar indicates 300 μm. Representative data obtained from two independent experiments are shown for hTSL$^{empty}$ (bulk) and hTSL$^{C19MC}$ (bulk) cells. Two hTSL$^{C19MC}$ clones were analyzed. See Supplementary Fig. 5a, b for detailed experimental design. **b** RNA-FISH of C19MC in hTSL$^{C19MC}$ and hTSL$^{empty}$ cells. C19MC expression was labeled with a Cy3-labeled oligonucleotide probe (shown in yellow). Nuclei were stained with Hoechst 33258 (shown in gray). The scale bar indicates 100 μm. **c** Percentage of the number of C19MC FISH signals per nucleus in hTSL$^{C19MC}$ and hTSL$^{empty}$ cells. Bulk hTSL$^{empty}$ and hTSL$^{C19MC}$ cells and two hTSL$^{C19MC}$ clones were analyzed. 500–1000 nuclei were examined for each cell type. **d** Relative expression levels of C19MC miRNAs in hTSL$^{C19MC}$ cells. Expression levels in hTS cells are set as 1. Two hTSL$^{C19MC}$ clones were analyzed. Bar charts are shown as mean. **e** DNA methylation analysis of the C19MC DMR by bisulfite sequencing. Black and white circles indicate methylated and unmethylated CpGs, respectively. The methylation levels are shown on the right. **f** Growth curve of hTSL$^{C19MC}$ clones and hTSL$^{empty}$ cells. Genetically unmodified hTS cells were also analyzed for comparison. Cells were cultured for 12 days. **g** Phase contrast and immunofluorescence images of EVT-like cells derived from hTSL$^{C19MC}$ and hTSL$^{empty}$ cells. Cells were stained for HLA-G. Nuclei were stained with Hoechst 33,258. The scale bar indicates 100 μm. Representative data obtained from two independent experiments are shown. Similar results were obtained for hTSL$^{C19MC}$ clones 1 and 2. **h** Flow cytometry analysis of HLA-G expression on EVT-like cells derived from hTSL$^{C19MC}$ clones. EVT-like cells derived from genetically unmodified hTS cells were analyzed as a positive control. **i** Phase contrast and immunofluorescence images of ST-like cells derived from hTSL$^{C19MC}$ and hTSL$^{empty}$ cells. Cells were stained for SDC1 and CGB. Nuclei were stained with Hoechst 33,258. The scale bar indicates 100 μm. Representative data obtained from two independent experiments are shown. Similar results were obtained for hTSL$^{C19MC}$ clones 1 and 2. **j** Fusion efficiency of ST-like cells derived from hTSL$^{C19MC}$ clones and hTSL$^{empty}$ cells. **k** Levels of hCG secreted by ST-like cells derived from hTSL$^{C19MC}$ clones and hTSL$^{empty}$ cells.

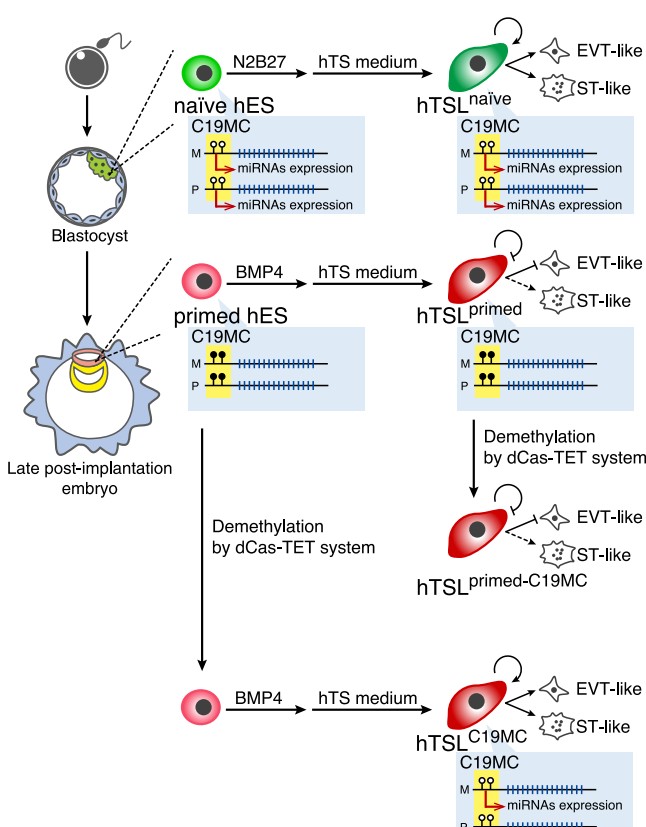

**Fig. 6 Summary of the derivation and characterization of hTSL cells.** Both alleles of the C19MC DMR are hypomethylated in naïve hES and hTSL$^{naïve}$ cells, but hypermethylated in primed hES and hTSL$^{primed}$ cells. The C19MC DMR is demethylated in hTSL$^{C19MC}$ and hTSL$^{primed-C19MC}$ cells using the dCas-TET system. Although hTSL$^{naïve}$ and hTSL$^{C19MC}$ cells are highly proliferative and can differentiate into both EVT- and ST-like cells, hTSL$^{primed}$ and hTSL$^{primed-C19MC}$ cells have poor proliferation and differentiation potentials. The C19MC DMR is highlighted in yellow. Black and white circles indicate methylated and unmethylated CpGs, respectively. M: maternal allele, P: paternal allele.

hES cells to hTS cells; (2) loss of C19MC expression can have irreversible effects on proliferation and differentiation of hTS cells. Our target prediction suggests that C19MC miRNAs might contribute to the derivation and maintenance of hTS cells by negatively regulating genes associated with the p53 signaling pathway. However, this prediction alone does not explain why the proliferation and differentiation potentials of C19MC-reactivated hTSL$^{primed}$ cells remains low. Further studies, including the narrowing down of functionally important miRNAs and conditional activation and inactivation of C19MC miRNAs, are needed to understand how C19MC works in hTS and hTSL cells. In addition, although we did not observe obvious abnormalities in hTSL$^{naïve}$ cells, it is still possible that excess C19MC miRNAs and other imprinting defects have unidentified adverse effects on these cells. Therefore, hTSL$^{naïve}$ cells may potentially be useful for investigating the roles of genomic imprinting in the human trophoblast lineage.

In conclusion, we identified C19MC as an important regulator of trophoblast proliferation and differentiation, as well as a critical determinant distinguishing human naïve and primed pluripotency. These findings are fundamental to understanding human trophoblast and epiblast development and provide a good example of epigenetic cell fate restriction during mammalian development.

## Methods

**Ethical considerations of working with human cells.** hTS cell lines #1-3 (TS$^{CT}$#1-3) were generated in our laboratory[6]. Primed hES cell lines (SEES1 and SEES4) were kindly provided by Drs. Hidenori Akutsu and Akihiro Umezawa (The National Center for Child Health and Development, Tokyo, Japan)[54]. All experimental protocols and procedures were approved by the Ethics Committee of Tohoku University Graduate School of Medicine (Research license 2017-1-814).

**Culture of hTS cells.** hTS cells were established in our previous study using hTS medium [DMEM/F12 (FUJIFILM Wako, Osaka, Japan; 048-29785) supplemented with 0.1 mM 2-mercaptoethanol (Thermo Fisher Scientific, Waltham, MA, USA; 21985023), 0.2% FBS (Thermo Fisher Scientific, 16141079), 0.5% Penicillin-Streptomycin (Thermo Fisher Scientific, 15140122), 0.3% BSA (FUJIFILM Wako, 017-22231), 1% ITS-X supplement (FUJIFILM Wako, 094-06761), 1.5 μg/ml L-ascorbic acid (FUJIFILM Wako, 013-12061), 50 ng/ml EGF (FUJIFILM Wako, 053-07871), 2 μM CHIR99021 (FUJIFILM Wako, 038-23101), 0.5 μM A83-01 (FUJIFILM Wako, 035-24113), 1 μM SB431542 (FUJIFILM Wako, 031-24291), 0.8 mM VPA (FUJIFILM Wako, 227-01071), and 5 μM Y27632 (FUJIFILM Wako, 257-00511)][6]. hTS cells were cultured on a plate coated with 7 μg/ml Col IV

(Corning, Corning, NY, USA; 354233) at 37 °C in 5% $CO_2$, and the culture medium was replaced every two days.

We recently modified hTS medium to improve single-cell cloning efficiency[55]. In this study, we maintained hTS cells in the modified hTS medium, which is composed of DMEM/F12 supplemented with 1% KSR (Thermo Fisher Scientific, 10828028), 0.5% Penicillin-Streptomycin, 0.15% BSA, 1% ITS-X supplement, 200 μM L-ascorbic acid, 50 ng/ml EGF, 2 μM CHIR99021, 5 μM A83-01, 0.8 mM VPA, and 2.5 μM Y-27632. hTS cells were cultured on a plate coated with 0.5 μg/ml iMatrix-511 (Nippi, Tokyo, Japan; 892011) at 37 °C in 5% $CO_2$, and the culture medium was replaced every two days. When hTS cells reached sub-confluence, they were dissociated using TrypLE Express (Thermo Fisher Scientific, 12604021) that was diluted with PBS at a 1:1 ratio. hTS cells were passaged every 4–6 days at a 1:20 split ratio.

### Culture of primed and naïve hES cells

Primed hES cells were maintained on Matrigel (Corning, 354234) in StemSure hPSC Medium Δ (FUJIFILM Wako, 197-17571) supplemented with 35 ng/ml FGF2 (FUJIFILM Wako, 064-05381), according to the manufacturer's protocol. 10 μM Y-27632 was added to the medium for 24 h after passage.

Naïve hES cells were generated from primed hES cells and cultured in 5i/L/A medium[14] as follows. We dissociated primed hES cells and seeded them on a mitomycin C-inactivated SNL feeder layer (Cell Biolabs, San Diego, CA, USA; CBA-316) in DMEM/F12 supplemented with 15% FBS, 5% KSR, 1 mM glutamine (Thermo Fisher Scientific, 25030-081), 1% non-essential amino acids (Thermo Fisher Scientific, 11140050), 0.5% Penicillin-Streptomycin, 0.1 mM 2-mercaptoethanol, 4 ng/ml FGF2, and 10 μM Y27632. After one day, the medium was switched to N2B27 medium [1:1 DMEM/F12 and Neurobasal (Thermo Fisher Scientific, 21103049) medium supplemented with 1% N2 supplement (Thermo Fisher Scientific, 17502048), 2% B27 supplement (Thermo Fisher Scientific, 17504044), 1% non-essential amino acids, 1 mM glutamine, 0.1 mM 2-mercaptoethanol, 0.5% Penicillin-Streptomycin, 50 μg/ml BSA (Sigma, St. Louis, MO, USA; A8806-1G)] supplemented with 5i/L/A [20 ng/ml LIF (FUJIFILM Wako, 125-06661), 20 ng/ml Activin A (FUJIFILM Wako, 014-23961), 1 μM PD0325901 (FUJIFILM Wako, 162-25291), 1 μM IM-12 (FUJIFILM Wako, 091-07131), 0.5 μM SB590885 (Adipogen Life Sciences, Liestal, Switzerland; SYN-1077), 1 μM WH-4-023 (FUJIFILM Wako, 234-02741), and 10 μM Y-27632]. Dome-shaped naïve hES cell colonies appeared within 10 days. Naïve hES cells were maintained on an SNL feeder layer in 5i/L/A medium under 5% $CO_2$ and 5% $O_2$ conditions. Naïve hES cells were passaged every 5–7 days by single-cell dissociation using 5 min treatment with Accutase (Innovative Cell Technologies, San Diego, CA, USA; AT104).

### Derivation of hTSL$^{naïve}$ and hTSL$^{primed}$ cells

Naïve hES cells were dissociated with Accutase and seeded on gelatin-coated plates (Sigma, G1890-100G) to remove SNL feeder cells. After an hour, the unattached naïve hES cells were cultured on iMatrix-511-coated plates in 5i/L/A medium. The next day, the medium was switched to N2B27 medium and the cells were cultured at 37 °C in 5% $CO_2$ and 20% $O_2$ conditions. The medium was replaced every 24 h. After three days of culture in N2B27 medium, cells were dissociated with TrypLE Express for 15 min at 37 °C. They were seeded on Col IV-coated plates at a 1:2 split ratio and cultured in hTS medium. hTSL cells were confirmed after a few passages.

Primed hES cells were seeded as single-cell monolayers (20,000 cells/cm$^2$) on Matrigel-coated plates with StemSure hPSC Medium Δ. After one day, the medium was changed to BMP4 medium [DMEM/F12 supplemented with 20% KSR, 1% Glutamax (Thermo Fisher Scientific, 35050-061), 1% non-essential amino acids, 0.1 mM 2-mercaptoethanol, and 1% Penicillin-Streptomycin supplemented with 50 ng/ml BMP4 (FUJIFILM Wako, 020-18851)][24]. Fresh BMP4 medium was applied every 24 h. After three days of culture in BMP4 medium, cells were dissociated with TrypLE Express for 15 min at 37 °C. They were seeded on a plate coated with 7 μg/ml Col IV at a 1:2 split ratio and cultured in hTS medium. Although the BMP4-treated primed hES cells were initially heterogeneous, hTSL cells were visible within a few passages.

After 5–9 passages, hTSL$^{naïve}$ and hTSL$^{primed}$ cells were transferred onto a plate coated with 0.5 μg/ml iMatrix-511 and cultured in the modified hTS medium. Unless otherwise noted, we used hTSL$^{naïve}$ and hTSL$^{primed}$ cells passaged over 11 times for this study.

### Differentiation of hTS, hTSL$^{naïve}$, and hTSL$^{primed}$ cells

Induction of EVT- and ST-like cells[6] was performed as follows. For the induction of EVT-hTS and -hTSL cells, hTS and hTSL cells were seeded at a density of 7500 cells/cm$^2$ on a plate coated with 1.5 μg/ml Col IV and cultured in EVT medium [DMEM/F12 supplemented with 0.1 mM 2-mercaptoethanol, 0.5% Penicillin-Streptomycin, 0.3% BSA, 1% ITS-X supplement, 50 ng/ml NRG1 (Cell Signaling, Danvers, MA, USA; 5218SC), 7.5 μM A83-01, 2.5 μM Y-27632, and 4% KSR]. Matrigel was added to the medium at a final concentration of 2% after the cells were seeded. On day 3, the medium was replaced with the EVT medium without NRG1, and Matrigel was added at a final concentration of 0.5%. The cells were analyzed on day 6.

For the induction of ST-hTS and ST-hTSL cells, hTS and hTSL cells were seeded at a density of 10,000 cells/cm$^2$ on a plate coated with 3 μg/ml Col IV and cultured in ST medium [DMEM/F12 supplemented with 0.1 mM 2-mercaptoethanol, 0.5% Penicillin-Streptomycin, 0.3% BSA, 1% ITS-X supplement, 2.5 μM Y-27632, 2 μM forskolin (FUJIFILM Wako, 067-02191), and 4% KSR]. The cells were analyzed on day 3.

### Deletion of the C19MC DMR

The gRNAs were designed using the Guide design tool (Zhang Lab; no longer available) or the Custom Alt-R CRISPR-Cas9 guide RNA design tool (IDT, Coralville, IA, USA). The lentiviral vector with inducible Cas9[55] was cotransfected with pCMV-VSV-G-RSV-Rev and pCAG-HIVgp (kindly provided by Dr. H. Miyoshi, RIKEN BioResource Center, Ibaraki, Japan) into 293 T cells (Takara Bio, Kusatsu, Japan; Z2180N) using Lipofectamine LTX (Thermo Fisher Scientific, 15338030). 10 μM Forskolin was added after 24 h of transfection. The supernatant was collected after 3 days of transfection, concentrated with Lenti-X Concentrator (Takara Bio, Z1231N), and used for the transduction of an hTS cell line (hTS#1) and a primed hES cell line (hES#1). Cas9 was induced via the addition of 50 ng/ml Doxycycline (Dox; Sigma, D9891-1G). After 24 h, a mixture of Lipofectamine RNAiMAX (Thermo Fisher Scientific, 13778030) and chemically synthesized gRNAs (IDT) was added to the cells. These cells were maintained in the modified hTS medium without Dox, and the medium was changed every 48 h. After single-cell cloning by limiting dilution, clones with deletion of the C19MC DMR were identified by PCR-based genetic screening. The parent-of-origin of the deleted allele(s) was determined using Sanger sequencing (Eurofins Genomics, Ebersberg, Germany). Sequences of the gRNAs and the primers used for PCR and Sanger sequencing are shown in Supplementary Data 4. SNPs distinguishing the parental alleles of the C19MC DMR were identified in our previous study[6].

### Demethylation of the C19MC DMR

The lentiviral vectors for the dCas-TET system were constructed as follows. The dCas9-peptide repeat and scFv-GFP-TET1 catalytic domain fusions (Addgene IDs: 82560 and 82561, Cambridge, MA, USA)[39] were PCR-amplified and cloned into the CS-CA-MCS plasmid (kindly provided by Dr. Hiroyuki Miyoshi, RIKEN BioResource Center, Ibaraki, Japan) using the In-Fusion HD Cloning kit (Takara Bio, Z9648N). The resulting vectors were designated as pCS-CA-dCas9-5xPlat2AflD and pCS-CA-scFvGCN4sf-GFP-TET1CD. A gRNA targeting the short tandem repeats within the C19MC DMR was designed and evaluated using CRISPRdirect[56] and cloned into the gRNA expression vector pCA-hU6[55] using the In-Fusion HD Cloning kit (Takara Bio). Sequences of the gRNA are shown in Supplementary Data 4.

Lentiviruses expressing dCas9, GFP-TET, and the gRNA were prepared as described above and transduced into a primed hES cell line (hES#1). Then, GFP-positive cells were sorted by flow cytometry with FACS Aria II (BD Biosciences, CA, USA) and used for the derivation of hTSL$^{C19MC}$ cells. We also transfected an hTSL$^{primed}$ cell line (hTSL$^{primed}$#1) with dCas9, GFP-TET, and the gRNA in the same way.

### Immunostaining and time-lapse imaging

For immunostaining, cells were fixed with 4% paraformaldehyde (PFA; FUJIFILM Wako) in PBS for 10 min, permeabilized with 0.3% Triton X-100 (FUJIFILM Wako) for 5 min, and blocked with 2% FBS/PBS for 30 min at room temperature. The cells were then incubated with primary antibodies overnight at 4 °C. The following primary antibodies were used: anti-NANOG (1:200), PE-conjugated anti-OCT4 (1:800), anti-GATA3 (1:800), anti-TFAP2C (1:200), anti-TP63 (1:100), anti-KLF17 (1:200), anti-ELF5 (1:100), anti-TEAD4 (1:100), anti-CDX2 (1:100), PE-conjugated anti-SDC1 (1:500), anti-hCG (1:10), and PE-conjugated anti-HLA-G (1:200) antibodies. Alexa Fluor 488-, 555-, or 647-conjugated anti-rabbit/mouse antibodies were used as secondary antibodies. Nuclei were stained with Hoechst 33258 (Dojin Chemicals, Kumamoto, Japan) and the images were taken with a fluorescence microscope (BZ-X710/810; Keyence, Osaka, Japan). Antibody information is provided in Supplementary Table 1.

Time-lapse images of hTS and hTSL cells differentiating into EVT-like cells were taken with BZ-X710. The cells were cultured in a temperature/$CO_2$ control chamber (STRG-KIW; TOKAI HIT, Shizuoka, Japan). Movies were created from the time-lapse images using BZ-X analyzer (Keyence).

### Flow cytometry

Cells were dissociated with TrypLE Express and suspended in 2% FBS/PBS. Fixation, permeabilization, and blocking were performed in the same way as immunostaining. For the analysis of KRT7, fixed cells were incubated with an anti-KRT7 antibody (1:100) overnight at 4 °C and stained with Alexa Fluor 488-conjugated anti-rabbit IgG for an hour. Normal rabbit IgG was used as an isotype control. For the analysis of HLA-ABC, unfixed cells were incubated with an FITC-conjugated anti-HLA-ABC (1:50) antibody for 15 min at room temperature. An FITC-conjugated mouse IgG2a was used as an isotype control. For the analysis of HLA-G, fixed cells were incubated with PE-conjugated anti-HLA-G (1:50) antibody overnight at 4 °C. A PE-conjugate mouse IgG1 antibody was used as an isotype control. Cells were passed through a 30 μm mesh filter just before flow cytometric analysis. Flow cytometry was carried out using FACS Canto II (BD Biosciences), and the data were analyzed using FlowJo software (BD Biosciences). Antibody information is provided in Supplementary Table 1.

**Measurement of hCG**. hTS, hTSL[primed], and hTSL[naïve] cells were differentiated into ST-like cells as described above. The supernatants were collected on day 3 and stored at −80 °C. The amount of secreted hCG was measured using hCG ELISA kit (Abnova, Taipei City, Taiwan; KA4005) according to the manufacturer's protocol. The absorbance was measured using a FlexStation 3 microplate reader (Molecular Devices, CA, USA).

**RNA fluorescence in situ hybridization (RNA-FISH) assay**. The RNA-FISH assay of C19MC was previously established[38,57] and performed as follows. Cells were fixed with 4% PFA/PBS for 20 min at room temperature and permeabilized with 70% ethanol for one day at 4 °C. Cells were hybridized overnight at 37 °C in the following solution: 15% formamide (FUJIFILM Wako, 066-02301), 2× SSPE (FUJIFILM Wako, 347-07545), 10% Dextran sulfate (FUJIFILM Wako, 194-13402), 150 μg/ml yeast tRNA (Thermo Fisher Scientific, 15401011), and 0.3 ng/μl of Alexa Fluor 488- or Cy3-labeled oligo-probes (Supplementary Data 4). Cells were washed at room temperature with 15% formamide/2× SSPE and 1× SSPE. Nuclei were stained with Hoechst 33258 (Dojin Chemicals, 343-07961), and the images were taken with a fluorescence microscope (BZ-X710). The oligo-probes target intronic sequences flanking C19MC miRNAs. Due to the repeated nature of the C19MC locus, these probes can hybridize to multiple locations along the C19MC pri-miRNA.

**Combined bisulfite restriction analysis (COBRA) and bisulfite sequencing (BS-seq)**. Genomic DNA was extracted with AllPrep DNA/RNA/miRNA Universal Kit (QIAGEN), treated with sodium bisulfite using EZ DNA Methylation-Gold Kit (Zymo Research, CA, USA), and amplified by PCR (TaKaRa EpiTaq HS). For COBRA of the C19MC DMR, the PCR products were digested with HhaI (NIPPON GENE, Toyama, Japan; 319-00162) and visualized on D1000 ScreenTape using the Agilent 2200 TapeStation instrument (Agilent Technologies). Undigested and digested fragments corresponded to unmethylated and methylated DNA, respectively. For BS-seq of the *ELF5* promoter and the C19MC DMR, PCR products were cloned into pGEM-T (Promega, WI, USA) and an average of 20 clones was sequenced using Sanger sequencing. Sequences of the primers used for PCR and Sanger sequencing are shown in Supplementary Data 4.

**RNA-seq**. Total RNA was extracted with AllPrep DNA/RNA/miRNA Universal Kit (QIAGEN, CA, USA; 80224) and genomic DNA was removed by digestion with RNase-free DNase I (QIAGEN, 79,254). RNA libraries were prepared using the TruSeq stranded mRNA LT Sample Prep Kit (Illumina, CA, USA) according to the manufacturer's protocol. RNA integrity was assessed using TapeStation 2200 (Agilent Technologies). All samples had an RNA integrity number equivalent (RINe) value of >9. The libraries were sequenced on the Illumina HiSeq 2500 platform (Illumina) with 101-bp paired-end reads. Sequenced data were first trimmed for quality control by TrimGalore (v0.6.5). The reads were aligned to the reference genome (UCSC hg38) using STAR (v2.7.3a)[58] with the Refseq gene annotation. Expression levels (TPM) of Refseq genes were calculated using RSEM[59]. Read counts were used to identify differentially expressed genes with the software DESeq2[60]. Transcripts less than 300-bp in length were excluded from the analyses.

**MiRNA-seq**. miRNA libraries were prepared using the QIAseq miRNA Library Kit (QIAGEN, 1103677) according to the manufacturer's protocol. RNA integrity was assessed using TapeStation 2200 (Agilent Technologies). All samples had an RINe value of >9. The libraries were sequenced on the Illumina HiSeq X Ten platform (Illumina) with 150-bp single-end reads. Sequenced data were first trimmed for quality control by fastp (v0.20.1)[61]. The trimmed reads were mapped to the hg38 reference genome using Bowtie 2 (v2.3.2)[62]. The mapped reads were annotated with miRBase (v22)[63] and counted using featureCounts (v1.6.4)[64]. Expression levels of mature miRNAs were normalized to reads per million (RPM).

**ChIP-Seq**. ChIP was performed on hTS cells using the ChIP Reagents (NIPPON GENE) and an anti-H3K4me3 antibody (MBL) according to the manufacturer's protocol. The ChIP-seq library was constructed using the Ovation Ultralow System V2 (NuGEN) and sequenced on the Illumina HiSeq 2500 platform (Illumina) with 101-bp paired-end reads. ChIP-seq reads were mapped to the hg19 reference genome using Bowtie 2 (v2.1.0)[62]. The genomic coordinates between hg19 and hg38 were converted using the LiftOver tool (https://genome.ucsc.edu/cgi-bin/hgLiftOver) provided by the USCS genome browser[65].

**WGBS**. WGBS was performed using the post-bisulfite adaptor-tagging (PBAT) method[66,67]. Briefly, genomic DNA was purified from hTSL[primed] cells with phenol/chloroform extraction and ethanol precipitation. Genomic DNA spiked with 0.5% (w/w) unmethylated lambda phage DNA (Promega) was used for library preparation according to the PBAT protocol. These DNA samples were treated with sodium bisulfite using EZ DNA Methylation-Gold Kit (Zymo Research). Concentrations of the PBAT products were quantified using the KAPA Library Quantification Kit for Illumina platforms (Kapa Biosystems). PBAT libraries were sequenced on the Illumina HiSeq 2500 (Illumina) with 101 bp single-end reads.

The reads were aligned to the reference genome (UCSC hg38) using Bismark (v0.19.1)[68]. The methylation level of each CpG site was calculated using the Bismark methylation extractor. We confirmed bisulfite conversion rates of >99% for all samples. For each CpG site, reads from both strands were combined to calculate the methylation level. Methylation levels of CpGs covered with ≥5 reads were analyzed.

**Annotations of genomic regions**. Annotations of Refseq genes and repeat sequences were downloaded from the UCSC Genome Browser[65]. Refseq transcripts shorter than 300 bp (encoding microRNAs or small nucleolar RNAs in most cases) were excluded from our analyses. Promoters were defined as regions 1,000 bp upstream and downstream from the transcription start sites of Refseq transcripts. Gene bodies were defined as transcribed regions of Refseq transcripts except for promoters. For calculation of the mean methylation levels, we analyzed genomic regions containing ≥5 CpGs with sufficient coverage for calculation of the methylation levels. The list of gDMRs in Supplementary Data 2 is from our previous study[32]. All of these gDMRs have been experimentally confirmed to show allele-specific DNA methylation and to be associated with imprinted genes[32,69]. In general, gDMRs are named after their closely associated imprinted genes. Similarly, we considered only gDMRs containing ≥5 CpGs.

**Functional enrichment pathway analysis of differentially expressed genes**. Functional annotation of differentially expressed genes was performed using the ConsensusPathDB human pathway database (http://cpdb.molgen.mpg.de/)[70].

**Prediction of target genes of miRNAs**. Potential target genes of miRNAs were predicted using mirDIP (http://ophid.utoronto.ca/mirDIP/)[40], which incorporates 30 different miRNA target databases. The top 1% of targets were considered.

**Graphical presentation**. Methylation levels of CpGs were visualized using Integrative Genomics Viewer (IGV) software (v2.8.0) (http://www.broadinstitute.org/igv/). Line charts, bar charts, contour plots, scatterplots, volcano plots, heatmaps, and circos plots were generated using the ggplot2, gplots, scatterplot3d, or circlize package in R (v3.1.3 and v4.1.2) (http://www.R-project.org/).

**Statistical analysis of the data**. All statistical analyses were performed using R, and a *p*- or *q*-value < 0.05 was considered statistically significant. The statistical methods used are described in the figure legends.

**Reporting summary**. Further information on research design is available in the Nature Research Reporting Summary linked to this article.

## Data availability
All sequencing data reported in this paper are deposited in DDBJ/GenBank/EMBL and Japanese Genotype-phenotype Archive (JGA) under the accession number DRA013428 and JGAS000107, respectively. Expression data of Refseq genes and miRNAs are provided with Supplementary Data 1 and 3, respectively. The output files of the Bismark methylation extractor for the methylation level of each CpG site in hTSL[primed] cells are deposited in DDBJ Genomic Expression Archive (GEA) under the accession number E-GEAD-474.

The WGBS data for hTS cells (JGA accession numbers: JGAS000107 and JGAS000112) were from our previous study[6]. We also included available WGBS data for naïve hES cells (Gene Expression Omnibus (GEO) accession numbers: GSM2041698 and GSM2041699)[33], hTSL[naïve] cells (GEO accession numbers: GSM4525520 and GSM4525521)[15], primed hES cells (GEO accession numbers: GSM706059 and GSM706060)[34].

ConsensusPathDB human pathway database (http://cpdb.molgen.mpg.de/)[70] for the functional enrichment pathway analysis and mirDIP (http://ophid.utoronto.ca/mirDIP/)[40] for predicting target genes of miRNAs were used.

Raw data of Figs. 2a, e, f, 3g, 4f, g, i, j, 5c, f, j, and k, and Supplementary Figs. 3c, 4c, and 5i are provided as a Source Data file.

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

## Acknowledgements

We would like to thank Dr. K. Nakayama, Dr. R. Funayama, Ms. M. Kikuchi, and Ms. A. Kitamura (Tohoku University) for technical assistance and Dr. RM. John (Cardiff University) for support and valuable suggestions. We thank Dr. H. Akutsu and Dr. A. Umezawa (National Center for Child Health and Development) for providing SEES1 and SEES4 hES cells. We also thank the Biomedical Research Core of Tohoku University Graduate School of Medicine for technical support. This work was supported by Japan Society for the Promotion of Science Grants-in-Aid for Scientific Research (JSPS KAKENHI) Grant 19K16135 and 22K15117 and the Sasakawa Scientific Research Grant from The Japan Science Society (to N. Kobayashi), and the Research Center for Biomedical Engineering (to H.K.), and La Ligue Régionale contre le Cancer (to J.C.), and KAKENHI Grant 19H05757 and 21H03072, Japan Agency for Medical Research and Development (AMED) Grant JP18bm0704021, and the Naito Foundation (to H.O.), and the Core Research for Evolutional Science and Technology from AMED Grants JP17gm0510011 and JP19gm1310001, KAKENHI Grant 17H04335 and 21H04834, Smoking Research Foundation, the Mitsubishi Foundation, and Takeda Science Foundation (to T.A.).

## Author contributions

N. Kobayashi performed most of the experiments. H.O. developed the culture conditions. N. Kobayashi, H.O., N. Kubota, C.K., and M.S. performed the informatics analyses. E.K., S.S., A.O., T.H., and H.K. were involved in refining the culture conditions, provided technical help and intellectual input, and edited the manuscript. H. Hiura, M.C., and J.C. provided experimental materials, technical help, intellectual input and critical feedback. H. Hamada provided intellectual input and revised the manuscript. N. Kobayashi, H.O., and T.A. conceived the study, planned experiments, interpreted the data, and wrote and revised the manuscript with input from all authors.

## Competing interests

The authors declare no competing interests.
