## [Peer Review File · Nature Communications]

The microRNA cluster C19MC confers differentiation potential into trophoblast lineages upon human pluripotent stem cellsREVIEWER COMMENTS

Reviewer #1 (Remarks to the Author):

This manuscript from the Arima laboratory reports that the epigenetic status of a primate-specific microRNA (miRNA) cluster located on chromosome 19 (C19MC) controls the potential of naïve and primed hES cells to differentiate into fully functional human trophoblast stem (hTS) cells. Over the past year a number of groups have reported that naïve hES cells have an increased potential to differentiate into hTS cells compared to their primed counterparts (Castel et al., 2020; Cinkornpumin et al., 2020; Dong et al., 2020; Guo et al., 2021; Io et al., 2021). Kobayashi et al. confirm these findings, but also show that hTS-like cells can be derived from the primed state by pre-treatment with BMP4 for three days. However, the resulting hTSL(primed) cells display a proliferation defect and fail to properly differentiate into terminally differentiated extravillous and syncytiotrophoblast cell types. By interrogating the DNA methylation landscape of hTSL(naïve) and hTSL(primed) cells, the authors noticed that the maternally imprinted C19MC DMR is hypermethylated in primed hES cells and hTSL(primed) cells, but demethylated in naïve hES cells and hTSL(naïve) cells. Accordingly, C19MC miRNAs are more highly expressed in naïve hES cells and hTSL(naïve) cells. The authors demonstrate that genetic ablation of the paternal copy of C19MC impairs the functionality of hTS cells previously established in their laboratory, while targeted demethylation of the C19MC in primed hES cells using a dCas9-TET system enables the derivation of hTSL(primed) cells with improved proliferation and differentiation competence.

Overall, this work is significant in that it presents a plausible molecular explanation for the divergent potential of naïve and primed hES cells to contribute to functional hTS cells. This is a highly topical question and these findings will be of interest to a broad readership. The experiments are well-designed and functionally significant, in particular the CRISPR deletion and dCas9-TET studies of the C19MC DMR. I have several suggestions for additional experiments and data presentation to further improve the rigor and clarity of this study:

Major comments:

1. Given the functional differences between naïve- and primed-derived hTS cells, it is somewhat surprising that the authors did not report significant transcriptomic differences. At what passage did they obtain these samples? If the proliferation of primed-derived hTS-like cells significantly slowed at later passages, they will likely observe a more dramatic transcriptomic difference then. Furthermore, the clustering of the samples in the heatmap in Fig. 3a suggests that there may be some differentially expressed genes between hTSL(primed) cells vs. hTSL(naïve) and hTS cells. It would be helpful to know what these differentially expressed genes are and to see how these samples cluster on a principal component analysis.
2. Deletion of the paternal C19MC DMR (or both the paternal and maternal DMRs) compromised the proliferation rate of hTS cells and their differentiation competence towards EVT and ST-like cells. However, the central argument of this manuscript is that human pluripotency is restricted by the silencing of C19MC. Therefore, it would be more convincing if the authors could perform CRISPR targeting in hES cells and demonstrate that homozygous deletion of the C19MC impairs the trophoblast potential of naïve hES cells.
3. The authors argued in text and diagram that demethylating the C19MC in primed hES cells "fully" restored their hTS cell potency, but ST-like cells derived from C19MC-reactivated primed hES cells secreted much less hCG than canonical and naïve-derived ST-like cells and did not show as good a morphology (Fig. 2d, f vs. Fig. 5i, k). This suggests that C19MC reactivation is not the sole determinant of trophoblast competence in human pluripotent cells. Can the authors include a transcriptional comparison of hTS-like and ST-like cells generated from C19MC-reactivated primed cells?
4. The model in Fig. 6 suggests that C19MC demethylation is a feature of naïve pluripotency associated with pre-implantation development, while primed cells corresponding to the post-

implantation epiblast undergo C19MC hypermethylation. Can the authors comment on whether a similar epigenetic silencing of the C19MC occurs during human embryogenesis in vivo? It may be possible to investigate this question using single cell bisulfite sequencing of human embryos cultured through implantation stages (Zhou et al., Nature, 2019).

5. The authors have randomly selected 5 miRNAs as a proxy for the expression of all 46 miRNAs within the C19MC cluster. It would be more representative and accurate to perform miRNA-seq.

Minor comments:

1. It is not clear whether the RNA FISH analysis in Fig. 3f and Fig. 4e targets a specific miRNA within the C19MC cluster, and if so, which one?

2. It should be clarified in Fig. 1A that hTS-like cells were derived from primed hES cells only after transient treatment with BMP4. While this information is included in the extended data, it would be helpful to indicate the requirement for BMP4 in Fig. 1A as well. At the moment this figure implies that hTS-like cells can be obtained by directly treating naïve and primed hES cells with hTS cell media.

Reviewer #2 (Remarks to the Author):

The authors identified that the primate-specific miRNA cluster C19MC was essential for hTS cell maintenance. They also found that reactivation of C19MC in primed hES cells could give rise to fully potent hTS cells. Thus, the authors claimed that C19MC was considered as a key cluster to distinguish the naïve and primed pluripotency. This work focused on the identification and functional analysis of C19MC, which was epigenetically regulated by DNA methylation, in hTS cell maintenance and differentiation. However, the exact mechanism research of C19MC especially these miRNAs was not enough for better understanding of the novel C19MC region. There are some important issues still need to be further addressed.

1. There are several protocols to generate naïve hES cells from primed hES cells (Chan Cell Stem Cell 2013, Gafni Nature 2013, Ware PNAS 2014, Duggal Stem Cells 2015), the authors may use at least 1-2 methods to induce naïve hES cells, to determine the effects of different naïve hES induction methods on cell differentiation and proliferation potentials of hTSLnaïve.

2. More trophoblast markers, including ELF5, TEAD4, and CDX2, needed to be tested in hTSLnaïve and hTSLprimed cells.

3. The authors claimed that they found a key miRNA cluster on chr.19 determined the phenotypic difference between naïve and primed hES cells. Though some evidences showed that epigenetic reactivation of C19MC would help primed hES cells to differentiate into hTS cells, how this region functioned on transdifferentiation and the roles of miRNAs in this region was unclear. Analysis of miRNA target genes in extended data or in the discussion is not enough for clearly determining the C19MC.

4. The authors claimed that they chose these five C19MC miRNAs arbitrarily, which confused me about the criterion. Besides, it was difficult to conclude that C19MC controlled the proliferation and differentiation of hTS cells. Whether the chosen five miRNAs functioned on the regulation of hTS proliferation or differentiation was still unknown.

5. The authors need to see whether deleted the C19MC DMR in naïve hES cells may affect the hTSLnaïve formation or the proliferation and differentiation potential of these hTSLnaïve cells(+/ Δ p).

6. Reactivation of C19MC in hES primed cells could give rise to the fully potent hTS cells, while reactivation of C19MC in hTS cells failed. If the C19MC activity was closely associated with the proliferation and differentiation potential of hTS cells, while the demethylation on C19MC promoter, relieved the silence of miRNA cluster, could not recover the hTS cell characteristics?

7. The author may test the gene expression profiles of hTS⁺/Δp, hTSΔm/Δp and hTSLC19MC, compare with the transcriptome of hTSLnaïve and hTSLprimed cells, which may be helpful to understand regulatory function/functional target genes of C19 microRNA cluster.

Reviewer #3 (Remarks to the Author):

The study by Kobayashi et al. identified the essential roles of C19MC in transdifferentiation of hESC to hTSCs based on the epigenetic and phenotypic comparisons between naive-type and primed-type hESCs. They were different in the ability to transdifferentiate into fully competent hTSCs. The experiments were well-designed and the interpretations of the data and the conclusions drawn are scientifically sound. I cannot find any critical fault throughout the paper. This study will provide invaluable information on the human TSC itself and the interchangeable relationship between hTSCs and hESCs, which are apparently different from those in mice.

I have several minor comments as follows:

1. Title and Abstract: I do not agree with the authors who stressed C19MC as a "key determinant" distinguishing human naive and primed pluripotency. Pluripotency defines the ability of stem cells to differentiate into all the tissues/cell types composing the body. Therefore, differentiation into TSCs is not included in this criterion. The point of this paper is that naive hESCs have a differentiation ability beyond pluripotency by the active C19MC.
2. Line 147: "The C19MC, KLHDC10, SLC4A7, and FAM50B DMRs met this criterion." Is this from Fig. 3c? If so, please indicate so here.
3. Line 190: I agree that there should be many putative targets of C19MC based on a computational prediction. However, it would be better to list a few examples that are known to be related to placental functions and showed upregulation upon C19MC null or paternal deletion.
4. Line 216: "one or two", not "more than one (= two or larger)", may be correct.
5. Line 237: "trophoblast cells in a broad sense" may be better.
6. Line 251: Please specify briefly what are different in the protocols of two studies.
7. Line 274: "that inhibit the derivation and maintenance of fully potent hTS cells." There is no data/evidence supporting this. In relation to the comment #3, the authors should explain this.
8. Fig. 3d: The data of the authors' hTSLnaïve seems lacking.
9. Fig. 3e,f: The expression levels of miRNAs and FISH-positive loci may indicate loss of imprinting of C19MC in hTSLnaïve. Were there any adverse effects of LOI on the phenotype of these TSCs??
10. Fig. 4e, f: The definition of the fusion index is shown here, but addition of a representative image (high mag. photo) will be helpful to the readers.

Point-by-point response to the reviewers' comments

Reviewer #1:

General comments:

This manuscript from the Arima laboratory reports that the epigenetic status of a primate-specific microRNA (miRNA) cluster located on chromosome 19 (C19MC) controls the potential of naïve and primed hES cells to differentiate into fully functional human trophoblast stem (hTS) cells. Over the past year a number of groups have reported that naïve hES cells have an increased potential to differentiate into hTS cells compared to their primed counterparts (Castel et al., 2020; Cinkornpumin et al., 2020; Dong et al., 2020; Guo et al., 2021; Io et al., 2021). Kobayashi et al. confirm these findings, but also show that hTS-like cells can be derived from the primed state by pre-treatment with BMP4 for three days. However, the resulting hTSL(primed) cells display a proliferation defect and fail to properly differentiate into terminally differentiated extravillous and syncytiotrophoblast cell types. By interrogating the DNA methylation landscape of hTSL(naïve) and hTSL(primed) cells, the authors noticed that the maternally imprinted C19MC DMR is hypermethylated in primed hES cells and hTSL(primed) cells, but demethylated in naïve hES cells and hTSL(naïve) cells. Accordingly, C19MC miRNAs are more highly expressed in naïve hES cells and hTSL(naïve) cells. The authors demonstrate that genetic ablation of the paternal copy of C19MC impairs the functionality of hTS cells previously established in their laboratory, while targeted demethylation of the C19MC in primed hES cells using a dCas9-TET system enables the derivation of hTSL(primed) cells with improved proliferation and differentiation competence.

Overall, this work is significant in that it presents a plausible molecular explanation for the divergent potential of naïve and primed hES cells to contribute to functional hTS cells. This is a highly topical question and these findings will be of interest to a broad readership. The experiments are well-designed and functionally significant, in particular the CRISPR deletion and dCas9-TET studies of the C19MC DMR. I have several suggestions for additional experiments and data presentation to further improve the rigor and clarity of this study:

Response: We greatly appreciate the reviewer's careful reading of our manuscript and the positive comments.

Major comment #1: Given the functional differences between naïve- and primed-derived hTS cells, it is somewhat surprising that the authors did not report significant transcriptomic differences. At what passage did they obtain these samples? If the proliferation of primed-derived hTS-like cells significantly slowed at later passages, they will likely observe a more dramatic transcriptomic difference then. Furthermore, the clustering of the samples in the heatmap in Fig. 3a suggests that there

may be some differentially expressed genes between hTSL(primed) cells vs. hTSL(naïve) and hTS cells. It would be helpful to know what these differentially expressed genes are and to see how these samples cluster on a principal component analysis.

Response: We thank the reviewer for the constructive suggestions. We conducted a principal component analysis and confirmed that the transcriptome profiles of hTS, hTSL^{primed}, and hTSL^{naïve} cells were similar but slightly different (Supplementary Fig. 2a). We also compared the transcriptome profiles of hTSL^{primed} cells with those of hTS and hTSL^{naïve} cells, which led to the identification of 274 down-regulated and 612 up-regulated genes (Supplementary Fig. 2c). Subsequent pathway enrichment analysis revealed that cell-cycle-related pathways were significantly enriched in the down-regulated genes (Supplementary Fig. 2d), which was consistent with the poor proliferation potential of hTSL^{primed} cells. We also found that pathways involved in cell-extracellular matrix (ECM) interactions were enriched in the up-regulated genes. It is well known that ECM plays crucial roles in EVT differentiation, and abnormal expression of ECM-related genes might provide an explanation on why hTSL^{primed} cells rarely differentiated into EVT-like cells. We have added these explanations to page 5, lines 127-137. We agree that a more dramatic transcriptomic difference might be observed when hTSL^{naïve} and hTSL^{primed} cells are analyzed at high passages. However, long-term culture can increase the risk of the accumulation of genetic and epigenetic changes, and we conducted RNA-seq of hTSL^{primed} and hTSL^{naïve} cells at relatively low passages (P12-19).

Supplementary Figure 2.

(a) Principal component analysis (PCA) of hES, hTS, and hTSL cells. Only 8,658 differentially expressed genes (TPM > 4 in at least one cell type, adjusted p-value < 0.05, fold change > 2) were analyzed. (c) Volcano plot comparing TSL^{primed} cells with hTS and hTSL^{naïve} cells. Gene expression levels in hTSL^{primed} cells were compared to those in hTS and hTSL^{naïve} cells, and expressed as log₂ fold change. The down- and up-regulated genes are shown in blue and red, respectively (TPM > 4 in at least one cell type, adjusted p-value < 0.05, fold change > 2). (d) Pathways enriched among the 274 down-regulated and 612 up-regulated genes (blue and red dots in Supplementary Fig. 2c). The top five pathways are shown with q-values.

Major comment #2: Deletion of the paternal C19MC DMR (or both the paternal and maternal DMRs) compromised the proliferation rate of hTS cells and their differentiation competence towards EVT and ST-like cells. However, the central argument of this manuscript is that human pluripotency is restricted by the silencing of C19MC. Therefore, it would be more convincing if the authors could perform CRISPR targeting in hES cells and demonstrate that homozygous deletion of the C19MC impairs the trophoblast potential of naïve hES cells.

Response: As suggested, we have tried to analyze whether homozygous deletion of C19MC impairs the trophoblast potential of naïve hES cells as follows. Since genetic manipulation in naïve hES cells is technically difficult, we first deleted the C19MC DMR in primed hES cells and isolated two heterozygous (hES^{+/ Δ}) and two homozygous (hES ^{Δ / Δ}) deletion clones (Supplementary Fig. 4d). Then, these cells were subjected to 5i/L/A medium to generate naïve hES cells. Naïve hES cells were successfully derived from the hES^{+/ Δ} clones but, to our surprise, not from the hES ^{Δ / Δ} clones (Supplementary Fig. 4e,f). This observation suggests that C19MC may be essential for naïve hES cell derivation from primed hES cells. We were therefore unable to directly address the trophoblast potential of naïve hES ^{Δ / Δ} cells. However, this new finding raises the interesting possibility that C19MC may be required for the transition from the primed state to the naïve state, and we have added this finding to our manuscript.

Supplementary Figure 4.

(a) The C19MC DMR was deleted in hES cells using two pairs of gRNAs (gRNA_deletion_pair1 and 2). Deletions were detected by genomic PCR. The position of three PCR primer sets is indicated. (d)

Confirmation of the deletion of the C19MC DMR in primed hES cells by genomic PCR. We obtained two heterozygous (hES^{+/ Δ}) and two homozygous (hES ^{Δ / Δ}) deletion clones. (e) Derivation of naïve hES cells from primed hES^{+/ Δ} and hES ^{Δ / Δ} cells using 5i/L/A medium. Naïve hES cells were successfully derived from primed hES^{+/ Δ} cells. However, primed hES ^{Δ / Δ} cells formed flattened colonies at P0, and these colonies failed to proliferate after passaging. Similar results were obtained with each of the two independent hES^{+/ Δ} and hES ^{Δ / Δ} clones. The scale bar indicates 100 μ m. (f) Immunostaining of a pluripotency marker (OCT4) and a naïve hES cell-specific marker (KLF17) in naïve hES^{+/ Δ} cells. Nuclei were stained with Hoechst 33258. The scale bar indicates 100 μ m. Similar results were obtained with two independent hES^{+/ Δ} clones.

Major comment #3: The authors argued in text and diagram that demethylating the C19MC in primed hES cells "fully" restored their hTS cell potency, but ST-like cells derived from C19MC-reactivated primed hES cells secreted much less hCG than canonical and naïve-derived ST-like cells and did not show as good a morphology (Fig. 2d, f vs. Fig. 5i, k). This suggests that C19MC reactivation is not the sole determinant of trophoblast competence in human pluripotent cells. Can the authors include a transcriptional comparison of hTS-like and ST-like cells generated from C19MC-reactivated primed cells?

Response: Thank you for raising this important issue. As suggested, we conducted additional RNA-seq of hTS-like and ST-like cells generated from C19MC-reactivated primed cells (hTSL^{C19MC} and ST-hTSL^{C19MC}). A principal component analysis revealed that hTSL^{C19MC} cells had transcriptome profiles similar to hTS and hTSL^{naïve} cells both before and after differentiation (Supplementary Fig. 5d). However, as the reviewer pointed out, ST-hTSL^{C19MC} cells secreted three to four times less hCG than canonical and naïve-derived ST-like cells. Thus, we further analyzed differentially expressed genes between ST-hTSL^{C19MC} cells vs. ST-hTS and ST-hTSL^{naïve} cells (Supplementary Fig. 5e). Subsequent pathway analysis revealed that cell-cycle-related genes were up-regulated in ST-hTSL^{C19MC} cells (Supplementary Fig. 5f), which might imply insufficient differentiation of these cells because mature ST cells are terminally differentiated and characterized by cell cycle arrest. Regarding the morphology, ST-hTSL^{C19MC} cells were similar to canonical and naïve-derived ST-like cells, but the original images in Fig. 5i were not clear enough. So, we have replaced the images of ST-hTSL^{C19MC} cells with clearer ones (Fig. 5i). From these data, we concluded that C19MC reactivation is the major, if not the sole, determinant of trophoblast competence in human pluripotent cells. We have added these explanations on pages 8-9, lines 246-259. Also, we have removed "fully potent" throughout the manuscript and referred to hTSL^{naïve} and hTSL^{C19MC} cells as "highly proliferative hTSL cells with the capacity to differentiate into EVT and ST cells" or "highly proliferative bipotent hTSL cells".

Supplementary Figure 5.

(d) Principal component analysis (PCA) of hTS and hTSL cells. hTSL^{C19MC} cells had transcriptome profiles similar to hTS and hTSL^{naïve} cells both before and after differentiation. Only 12,853 differentially expressed genes (TPM > 4 in at least one cell type, adjusted p-value < 0.05, fold change > 2) were analyzed. (e) Volcano plot comparing ST-hTSL^{C19MC} cells with ST-hTS and -hTSL^{naïve} cells. Gene expression levels were compared and expressed as log₂ fold change. The down- and up-regulated genes are shown in blue and red dots, respectively (TPM > 4 in at least one cell type, adjusted p-value < 0.05, fold change > 2). (f) Pathways enriched among the genes down- and up-regulated in ST-hTSL^{C19MC} cells (blue and red dots in Supplementary Fig. 5e). The top five pathways of the down- and up-regulated genes are represented with q-values on the left and right charts, respectively.

Figure 5.

Major comment #4: The model in Fig. 6 suggests that C19MC demethylation is a feature of naïve pluripotency associated with pre-implantation development, while primed cells corresponding to the post-implantation epiblast undergo C19MC hypermethylation. Can the authors comment on whether a similar epigenetic silencing of the C19MC occurs during human embryogenesis in vivo? It may be possible to investigate this question using single cell bisulfite sequencing of human embryos cultured through implantation stages (Zhou et al., Nature, 2019).

Response: Thank you for the insightful suggestion. We reanalyzed the single-cell WGBS (scWGBS) data of human epiblast cells (Zhou et al., Nature, 2019). Please note that Zhou et al. conducted scWGBS only on Day 6, 8, and 10 embryos. Our analysis revealed that the C19MC DMR maintains intermediate methylation levels at all the analyzed time points, as shown in the following Fig-A. Moreover, most of the sequence reads mapped to the C19MC DMR were hypo- (<20%) or hypermethylated (>80%), implying that the C19MC DMR may still be imprinted in the epiblast lineage until at least day 10 post-fertilization (Fig-B). Thus, unfortunately, it was unclear when the post-implantation epiblast cells undergo C19MC hypermethylation. Since these data are inconclusive, we have decided not to include them in the revised manuscript. Instead, we have added the following sentence (page 10, lines 323-324): “it remains unknown when the C19MC DMR undergoes hypermethylation in the epiblast lineage during human development”.

This Figure is for review only.

(A) DNA methylation patterns at the C19MC locus in hTS, hES, TE, ICM, and post-implantation epiblast cells. The C19MC DMR is highlighted in yellow and its methylation levels are indicated. (B) Analysis of individual sequence reads mapped to the C19MC DMR. Sequence reads were classified into five groups according to their methylation levels. The distribution of the methylation levels is shown as stacked bar charts. The scWGBS data of human ICM, TE (Zhu et al., Nat Genet., 2018), and epiblast cells (Zhou et al., Nature, 2019) were merged and analyzed.

Major comment #5: The authors have randomly selected 5 miRNAs as a proxy for the expression of all 46 miRNAs within the C19MC cluster. It would be more representative and accurate to perform miRNA-seq.

Response: As suggested, we conducted miRNA-seq of hES, hTS, and hTSL cells (Fig. 3e, 4d, and 5d). We divided the C19MC miRNAs into two classes based on their location. Class I miRNAs are located between the C19MC DMR and the putative second promoter. Class II miRNAs are downstream of the putative second promoter (please see Fig. 3a for details). This miRNA-seq data analysis reinforced our claim that Class I miRNAs were active in hTS, hTSL^{naive}, and hTSL^{C19MC} cells but silenced in hTSL^{primed} and C19MC KO hTS cells. As described in our original manuscript, Class II miRNAs were expressed in hTSL^{primed} and C19MC KO hTS cells, and their regulatory mechanism may be different from that of Class I miRNAs.

Figure 3.

(e) Relative expression levels of C19MC miRNAs in hES, hTS, and hTSL cells. Expression levels in hTS cells are set as 1. Data are shown as mean + SE. Note that hsa-mir-1283 and -521 have copies in both the Class I and II regions.

Figure 4.

(d) Relative expression levels of C19MC miRNAs in C19MC knockout hTS cells. Expression levels in hTS cells are set as 1. Data are shown as mean + SE.

Figure 5.

(d) Relative expression levels of C19MC miRNAs in hTSL^{C19MC} cells. Expression levels in hTS cells are set as 1. Data are shown as mean + SE.

Minor comment #1: It is not clear whether the RNA FISH analysis in Fig. 3f and Fig. 4e targets a specific miRNA within the C19MC cluster, and if so, which one?

Response: We are sorry for the lack of information. We used a mixture of two RNA-FISH probes (Supplementary Data 4). These probes target intronic sequences flanking C19MC miRNAs, not specific miRNAs. Due to the repeated nature of the C19MC locus, these probes can hybridize to multiple locations along the C19MC pri-miRNA. Thus, these DNA oligo-probes detect nuclear pri-miRNA species nearby their transcription sites (Clément et al., J Cell Sci. 2012), and not fully processed cytoplasmic miRNAs.

Probe	Oligo sequences
Probe 1	AXTTTCCTTGACCAGGXTAAAATGGACACAAAAAXAAAGATGCATTXA
Probe 2	GAXGATAAGCATGCXCTGCAAATGTATXACCAAGATCAGCATCAACXT

X: Amino-allyl-modified nucleotides. X is labeled with a fluorescent dye (Alex448 or Cy5).

Minor comment #2: It should be clarified in Fig. 1A that hTS-like cells were derived from primed hES cells only after transient treatment with BMP4. While this information is included the extended data, it would be helpful to indicate the requirement for BMP4 in Fig. 1A as well. At the moment this figure implies that hTS-like cells can be obtained by directly treating naïve and primed hES cells with hTS cell media.

Response: We have added a schematic representation of the derivation of hTSL cells in Fig. 1a to indicate that hTS-like cells were derived from primed hES cells only after transient treatment with BMP4.

Figure 1.
a

(a) Phase-contrast images of hTSL cells derived from naïve and primed hES cells. Naïve hES cells were derived from primed hES cells using 5i/L/A medium. After three days of culture in N2B27 medium, naïve hES cells were dissociated and cultured in hTS medium. Primed hES cells were treated with BMP4 for three days and then, the resultant cells were dissociated and cultured in hTS medium. hTSL cells were visible within a few passages (white dotted line). Similar results were obtained with two independent hES cell lines. An image of hTS cells (hTS#1) is shown for comparison. The scale bar indicates 300 μm .

Reviewer #2:

General comments:

The authors identified that the primate-specific miRNA cluster C19MC was essential for hTS cell maintenance. They also found that reactivation of C19MC in primed hES cells could give rise to fully potent hTS cells. Thus, the authors claimed that C19MC was considered as a key cluster to distinguish the naïve and primed pluripotency. This work focused on the identification and functional analysis of C19MC, which was epigenetically regulated by DNA methylation, in hTS cell maintenance and differentiation. However, the exact mechanism research of C19MC especially these miRNAs was not enough for better understanding of the novel C19MC region. There are some important issues still need to be further addressed.

Response: We greatly appreciate the reviewer's critical reading of our manuscript and the constructive comments.

Comment #1: There are several protocols to generate naïve hES cells from primed hES cells (Chan Cell Stem Cell 2013, Gafni Nature 2013, Ware PNAS 2014, Duggal Stem Cells 2015), the authors may use at least 1-2 methods to induce naïve hES cells, to determine the effects of different naïve hES induction methods on cell differentiation and proliferation potentials of hTSLnaïve.

Response: Several groups have recently reported the derivation of hTS-like cells from naïve hES cells, and the proliferation and differentiation potentials of these hTS-like cells were comparable to conventional hTS cells. In these studies, three different protocols (5i/L/A, t2iLGö, and PXGL media) have been applied to generate naïve hES cells, and all the protocols worked similarly well with consistent findings, demonstrating reproducibility irrespective of the different protocols. Importantly, in our current study, hTS cell derivation from naïve hES cells was used only as a positive control. Therefore, rather than adding new data, we would like to refer to the above-mentioned studies as follows (page 3, lines 53-54): "Recent studies reveal that naïve human ES (hES) cells derived using three different protocols¹²⁻¹⁴ can spontaneously differentiate into trophoblast stem-like (hTSL) cells¹⁵⁻¹⁸. These hTSL cells have similar proliferation and differentiation capacities and transcriptome and methylome profiles to human TS (hTS) cells."

Comment #2: More trophoblast markers, including ELF5, TEAD4, and CDX2, needed to be tested in hTSLnaïve and hTSLprimed cells.

Response: As suggested, we performed immunostaining of ELF5, TEAD4, and CDX2 (Supplementary Fig.1d). We confirmed that ELF5 and TEAD4 were expressed in hTSL^{naïve} and hTSL^{primed} cells. Please note that CDX2 is not a marker of hTS cells, and this protein was undetectable in hTSL^{naïve} and hTSL^{primed} cells as in the case of hTS cells.

Supplementary Figure 1.

Comment #3: The authors claimed that they found a key miRNA cluster on chr.19 determined the phenotypic difference between naïve and primed hES cells. Though some evidences showed that epigenetic reactivation of C19MC would help primed hES cells to differentiate into hTS cells, how this region functioned on transdifferentiation and the roles of miRNAs in this region was unclear. Analysis of miRNA target genes in extended data or in the discussion is not enough for clearly determining the C19MC.

Response: Thank you for the valuable comment. We agree that it is important to understand how C19MC functions. However, it is extremely difficult to experimentally address this issue because C19MC contains 46 miRNAs and there should be many target genes. Therefore, we decided to combine computational target prediction and RNA-seq data to identify the most likely targets and estimate the function of C19MC as follows. We compared the RNA-seq data of C19MC active (hTS, hTSL^{naïve}, hTS ^{Δ m/+}, and hTSL^{C19MC}) and inactive (hTSL^{primed}, hTS^{+/ Δ p}, hTS ^{Δ m/ Δ p}, and hTSL^{control}) cells and identified genes that were up-regulated in the C19MC inactive cells. By merging these up-regulated genes with computationally predicted target genes, we identified 310 genes as the most likely candidates (Supplementary Fig. 6b). Subsequent pathway enrichment analysis revealed that p53-

related pathways were significantly enriched among these 310 genes (Supplementary Fig. 6c), which is consistent with the poor proliferation potential of the C19MC inactive cells. We further looked at the combination between the p53 signaling-associated genes and C19MC miRNAs and found that some negative regulators of cell growth, such as *CCNG2*, *CDKN1A*, *PMAIP1*, *TP53INP1*, and *ZMAT3*, were potentially targeted by multiple C19MC miRNAs (Supplementary Fig. 6d).

Supplementary Figure 6.

(b) Venn diagram showing the most likely target genes of the Class I miRNAs (*please see Fig.3d for the classification of the C19MC miRNAs*). The gene expression levels in C19MC inactive (hTSL^{primed}, hTSL^{+/ Δ p}, hTSL ^{Δ m/ Δ p}, and hTSL^{control}) cells were compared with those in C19MC active (hTS, hTSL^{naive}, hTSL ^{Δ m/+} and hTSL^{C19MC}) cells, and 1,030 up-regulated genes were defined as follows: TPM > 4 in at least one cell type, adjusted p-value < 0.05, and fold change > 1.5. By merging these up-regulated genes with the computationally predicted target genes in (a), we identified 310 genes as the most likely candidates. (c) Overrepresented pathways among the most likely target genes of the Class I miRNAs. The top five pathways are represented with q-values. (d) Circos plot of the miRNA-mRNA combinations associated with “p53 transcriptional gene network” and “p53 signaling pathway”. The expression levels in C19MC active cells were compared to those in C19MC inactive cells and expressed as the log2 fold change.

Comment #4: The authors claimed that they chose these five C19MC miRNAs arbitrarily, which confused me about the criterion. Besides, it was difficult to conclude that C19MC controlled the proliferation and differentiation of hTS cells. Whether the chosen five miRNAs functioned on the

regulation of hTS proliferation or differentiation was still unknown.

Response: We are sorry for the confusion. To analyze C19MC miRNAs in an unbiased way, we have conducted miRNA-seq in the revised manuscript (Fig. 3e, 4d, and 5d). We divided the C19MC miRNAs into two classes based on their location. Class I miRNAs are located between the C19MC DMR and the putative second promoter. Class II miRNAs are downstream of the putative second promoter (please see Fig. 3d for details). This miRNA-seq data analysis reinforced our claim that Class I miRNAs were active in hTS, hTSL^{naïve}, and hTSL^{C19MC} cells but silenced in hTSL^{primed} and C19MC KO hTS cells. As described in our original manuscript, Class II miRNAs were expressed in hTSL^{primed} and C19MC KO hTS cells, and their regulatory mechanism may be different from that of Class I miRNAs. We have discussed the putative function of C19MC miRNAs in the response to Comment #3 of this reviewer.

Figure 3.

(e) Relative expression levels of C19MC miRNAs in hES, hTS, and hTSL cells. Expression levels in hTS cells are set as 1. Data are shown as mean + SE. Note that Hsa-mir-1283 and -521 have copies in both the Class I and II regions.

Figure 4.

(d) Relative expression levels of C19MC miRNAs in C19MC knockout hTS cells. Expression levels in hTS cells are set as 1. Data are shown as mean + SE.

Figure 5.

(d) Relative expression levels of C19MC miRNAs in hTSL^{C19MC} cells. Expression levels in hTS cells are set as 1. Data are shown as mean + SE.

Comment #5: The authors need to see whether deleted the C19MC DMR in naïve hES cells may affect the hTSLnaïve formation or the proliferation and differentiation potential of these hTSLnaïve cells(+/ Δ p).

Response: As suggested, we have tried to generate naïve hES cells with the deletion of the C19MC DMR as follows. Since genetic manipulation in naïve hES cells is technically difficult, we first deleted the C19MC DMR in primed hES cells and isolated two heterozygous (hES^{+/ Δ}) and two homozygous (hES ^{Δ / Δ}) deletion clones (Supplementary Fig. 4d). Then, these cells were subjected to 5i/L/A medium to generate naïve hES cells. Naïve hES cells were successfully derived from the hES^{+/ Δ} clones but, to our surprise, not from the hES ^{Δ / Δ} clones (Supplementary Fig. 4e,f). This observation suggests that C19MC may be essential for naïve hES cell derivation. We were therefore unable to directly address the trophoblast potential of naïve hES ^{Δ / Δ} cells. However, this new finding raises the interesting possibility that C19MC may be required for the transition from the primed state to the naïve state, and we have added this finding to our manuscript. Please note that C19MC is biallelically expressed in naïve hES cells, and we needed to delete both alleles to generate C19MC-deficient naïve hES cells.

Supplementary Figure 4.

(a) The C19MC DMR was deleted in hTS and hES cells using two pairs of gRNAs (gRNA_deletion_pair1 and 2). Deletions were detected by genomic PCR. The position of three PCR primer sets is indicated. (d) Confirmation of the deletion of the C19MC DMR in primed ES cells by genomic PCR. The C19MC DMR was deleted in primed hES cells using two pairs of gRNAs (gRNA_deletion_pair1 and 2). We obtained two heterozygous (hES^{+/ Δ}) and two homozygous (hES ^{Δ / Δ})

deletion clones. (e) Derivation of naïve hES cells from primed hES^{+Δ} and hES^{ΔΔ} cells using 5i/L/A medium. Naïve hES cells were successfully derived from primed hES^{+Δ} cells. However, primed hES^{ΔΔ} cells formed flattened colonies at P0, and these colonies failed to proliferate after passaging. Similar results were obtained with each of the two independent hES^{+Δ} and hES^{ΔΔ} clones. The scale bar indicates 100 μm. (f) Immunostaining of a pluripotency marker (OCT4) and a naïve hES cell-specific marker (KLF17) in naïve hES^{+Δ} cells. Nuclei were stained with Hoechst 33258. The scale bar indicates 100 μm. Similar results were obtained with two independent hES^{+Δ} clones.

Comment #6: Reactivation of C19MC in hES primed cells could give rise to the fully potent hTS cells, while reactivation of C19MC in hTS cells failed. If the C19MC activity was closely associated with the proliferation and differentiation potential of hTS cells, while the demethylation on C19MC promoter, relieved the silence of miRNA cluster, could not recover the hTS cell characteristics?

Response: As pointed out, fully potent hTS cells can be derived from C19MC-reactivated primed hES cells but not from C19MC-reactivated hTSL^{primed} cells. This result was unexpected for us too. We think there are at least two possibilities to explain this discrepancy: (1) C19MC might be required not only for the maintenance of hTS cells but also for the transition from primed hES cells to hTS cells; (2) loss of C19MC expression can have irreversible effects on proliferation and differentiation of hTS cells. We have referred to these possibilities in the discussion (page 11, lines 328-332).

Comment #7: The author may test the gene expression profiles of hTS^{+Δp}, hTS^{Δm/Δp} and hTSLC19MC, compare with the transcriptome of hTSL^{naïve} and hTSL^{primed} cells, which may be helpful to understand regulatory function/functional target genes of C19 microRNA cluster.

Response: Thank you for the constructive suggestion. As described in the response to Comment #3 of this reviewer, we combined computational target prediction and RNA-seq data to identify the most likely targets of C19MC miRNAs and estimated their function.

Reviewer #3:

General comments: The study by Kobayshi et al. identified the essential roles of C19MC in transdifferentiation of hESC to hTSCs based on the epigenetic and phenotypic comparisons between naive-type and primed-type hESCs. They were different in the ability to transdifferentiate into fully competent hTSCs. The experiments were well-designed and the interpretations of the data and the conclusions drawn are scientifically sound. I cannot find any critical fault throughout the paper. This study will provide invaluable information on the human TSC itself and the interchangeable relationship between hTSCs and hESCs, which are apparently different from those in mice.

Response: We are grateful for the reviewer's appreciation of our work and the positive comments.

Minor comment #1: Title and Abstract: I do not agree with the authors who stressed C19MC as a "key determinant" distinguishing human naive and primed pluripotency. Pluripotency defines the ability of stem cells to differentiate into all the tissues/cell types composing the body. Therefore, differentiation into TSCs is not included in this criterion. The point of this paper is that naive hESCs have a differentiation ability beyond pluripotency by the active C19MC.

Response: As suggested, we have revised the title and abstract as follows:

[Title]

"Activation of the chromosome 19 miRNA cluster confers differentiation potential into trophoblast lineages upon human pluripotent stem cells"

[Abstract]

Original: "we identify C19MC as a key determinant distinguishing human naïve and primed pluripotency"

Revised: "we reveal that C19MC activation confers differentiation potential into trophoblast lineages on hES cells."

Minor comment #2: Line 147: "The C19MC, KLHDC10, SLC4A7, and FAM50B DMRs met this criterion." Is this from Fig. 3c? If so, please indicate so here.

Response: Thank you for pointing this out. We have indicated that the description is based on Fig. 3c (page 6, line 163).

Minor comment #3: Line 190: I agree that there should be many putative targets of C19MC based on a computational prediction. However, it would be better to list a few examples that are known to be related to placental functions and showed up-regulation upon C19MC null or paternal deletion.

Response: Thank you for the valuable suggestion. We combined computational target prediction and RNA-seq data to identify the most likely targets of C19MC miRNAs and estimated their function as follows. We compared the RNA-seq data of C19MC active (hTS, hTSL^{naive}, hTSL^{Δm/+}, hTSL^{C19MC}) and inactive (hTSL^{primed}, hTS^{+Δp}, hTS^{Δm/Δp}, and hTSL^{control}) cells and identified genes that were up-regulated in the C19MC inactive cells. By merging these up-regulated genes with computationally predicted target genes, we identified 310 genes as the most likely candidates (Supplementary Fig. 6b). Subsequent pathway enrichment analysis revealed that p53-related pathways were significantly enriched among these 310 genes (Supplementary Fig. 6c), which is consistent with the poor proliferation potential of the C19MC inactive cells. We further looked at the combination between the p53 signaling-associated genes and C19MC miRNAs and found that some negative regulators of cell growth, such as *CCNG2*, *CDKN1A*, *PMAIP1*, *TP53INP1*, and *ZMAT3*, were potentially targeted by multiple C19MC miRNAs (Supplementary Fig. 6d).

Supplementary Figure 6.

(b) Venn diagram showing the most likely target genes of the Class I miRNAs (please see Fig.3d for the classification of the C19MC miRNAs). The gene expression levels in C19MC inactive (hTSL^{primed}, hTS^{+Δp}, hTS^{Δm/Δp}, and hTSL^{control}) cells were compared with those in C19MC active (hTS, hTSL^{naive}, hTS^{Δm/+} and hTSL^{C19MC}) cells, and 1,030 up-regulated genes were defined as follows: TPM > 4 in at least one cell type, adjusted p-value < 0.05, and fold change > 1.5. By merging these up-regulated

genes with the computationally predicted target genes in (a), we identified 310 genes as the most likely candidates. (c) Overrepresented pathways among the most likely target genes of the Class I miRNAs. The top five pathways are represented with q-values. (d) Circos plot of the miRNA-mRNA combinations associated with “p53 transcriptional gene network” and “p53 signaling pathway”. The expression levels in C19MC active cells were compared to those in C19MC inactive cells and expressed as the log₂ fold change.

Minor comment #4: Line 216: “one or two”, not “more than one (= two or larger)”, may be correct.

Response: We have amended the phrase as suggested (page 8, line 240).

Minor comment #5: Line 237: “trophoblast cells in a broad sense” may be better.

Response: We have amended the phrase as suggested (page 10, line 293).

Minor comment #6: Line 251: Please specify briefly what are different in the protocols of two studies.

Response: We have included the following explanation in the discussion (page 10, lines 307-309): “Li et al. generated trophoblast cysts using a micromesh culture technique. Mischler et al. induced and expanded trophoblast cells using media containing sphingosine-1 phosphate or its agonist.”

Minor comment #7: Line 274: “that inhibit the derivation and maintenance of fully potent hTS cells.” There is no data/evidence supporting this. In relation to the comment #3, the authors should explain this.

Response: Thank you for pointing this out. As described in the response to Comment #3 of this reviewer, p53-related pathways were enriched among the predicted target genes of C19MC miRNAs. Based on this result, we revised the sentence as follows (page 11, lines 332-334):

Original: “We infer that C19MC miRNAs may negatively regulate target mRNAs that inhibit the derivation and maintenance of fully potent hTS cells”

Revised: “Our target prediction suggests that C19MC miRNAs might contribute to the derivation and maintenance of hTS cells by negatively regulating genes associated with the p53 signaling pathway.”

Minor comment #8: Fig. 3d: The data of the authors' hTSLnaive seems lacking.

Response: We are sorry for the confusion. The DNA methylation data of our hTSL^{naive} cells, which were obtained by conventional bisulfite sequencing, not by WGBS, are in Supplementary Fig. 3b. To avoid any confusion, we have added the following sentence in the legend of Fig. 3d: "Please see Supplementary Fig. 3b for the DNA methylation patterns of the C19MC DMR in hTSL^{naive} cells derived in this study".

Minor comment #9: Fig. 3e,f: The expression levels of miRNAs and FISH-positive loci may indicate loss of imprinting of C19MC in hTSLnaive. Were there any adverse effects of LOI on the phenotype of these TSCs??

Response: Thank you for the insightful comment. Although we did not observe obvious abnormalities in hTSL^{naive} cells, it is still possible that excess C19MC miRNAs have unidentified adverse effects on these cells. We have referred to this possibility in the revised manuscript as follows (page 11, lines 338-341): "although we did not observe obvious abnormalities in hTSL^{naive} cells, it is still possible that excess C19MC miRNAs and other imprinting defects have unidentified adverse effects on these cells. Therefore, hTSL^{naive} cells may potentially be useful for investigating the roles of genomic imprinting in the human trophoblast lineage."

Minor comment #10: Fig. 4e, f: The definition of the fusion index is shown here, but addition of a representative image (high mag. photo) will be helpful to the readers.

Response: As suggested, we have added high magnification images in Fig. 4h.

REVIEWERS' COMMENTS

Reviewer #1 (Remarks to the Author):

The authors fully addressed my questions and have done a commendable job of revising their manuscript. They have added new experimental data suggesting that the C19MC may be required for the primed-to-naïve transition as well as for generation of functional hTSCs. In addition, they included a computational miRNA target prediction, which indicates that the function of the C19MC in hTSCs is to repress pathways related to p53 signaling and extracellular matrix interactions. The revised manuscript provides important mechanistic insight into the trophoblast potential of naïve and primed hESCs and will be of substantial interest for the readers of Nature Communications.

Minor comments:

1. On lines 275-276, the authors mention that "hTSL-primed" and "hTSL-control" cells were used to identify the predicted target genes of Class I miRNAs that were upregulated in C19MC inactive cells. The meaning of the term "hTSL-control" cells here is unclear since hTSL cells generated from naïve hESCs express C19MC. Are "hTSL-control" cells similar to the bulk "hTSL-empty" cells described in Fig. 5? Please clearly define this term, which is also mentioned but not explained in the legend for Supplementary Figure 6a.

2. In Fig. 6 it may be helpful to indicate "Demethylation by dCas-TET". At the moment it is not clear from the bottom of this figure how demethylation was achieved and the cartoon could be misinterpreted to suggest that demethylation is a consequence of BMP4 treatment.

Reviewer #2 (Remarks to the Author):

The authors have addressed most of my comments, and together with the incorporation of the suggestions of the other reviewers, the manuscript is substantially improved. I feel this manuscript is suitable for publication in Nature Communications now.

Reviewer #3 (Remarks to the Author):

The authors have thoughtfully addressed all previous comments and suggestions. The paper is now more clear and is even stronger than the first version.

Point-by-point response to the reviewers' comments

Reviewer #1:

Minor comment #1: On lines 275-276, the authors mention that ‘hTSL-primed’ and “hTSL-control” cells were used to identify the predicted target genes of Class I miRNAs that were upregulated in C19MC inactive cells. The meaning of the term “hTSL-control” cells here is unclear since hTSL cells generated from naïve hESCs express C19MC. Are “hTSL-control” cells similar to the bulk “hTSL-empty” cells described in Fig. 5? Please clearly define this term, which is also mentioned but not explained in the legend for Supplementary Figure 6a.

Response: We are sorry for the mistake. “hTSL-control” is a typo, and we have replaced it with “hTSL^{empty}”.

Minor comment #2: In Fig. 6 it may be helpful to indicate “Demethylation by dCas-TET”. At the moment it is not clear from the bottom of this figure how demethylation was achieved and the cartoon could be misinterpreted to suggest that demethylation is a consequence of BMP4 treatment.

Response: We have revised Fig. 6 as suggested.